# Eco-hydrological effects of stream-aquifer water interaction: A case study of the Heihe River Basin, northwestern China

Yujin Zeng[1, 2], Zhenghui Xie[1], Yan Yu[3], Shuang Liu[1, 2], Linying Wang[1, 2], Binghao Jia[1], Peihua Qin[1], Yaning Chen[4]

[1]State Key Laboratory of Numerical Modeling for Atmospheric Sciences and Geophysical Fluid Dynamics, Institute of Atmospheric Physics, Chinese Academy of Sciences, Beijing, 100029, China
[2]College of Earth Science, University of Chinese Academy of Sciences, Beijing 100049, China
[3]Zhejiang Institute of Meteorological Sciences, Hangzhou, 310008, China
[4]Key Laboratory of Oasis Ecology and Desert Environment, Xinjiang Institute of Ecology and Geography, Chinese
Academy of Sciences, Urumqi, 830011, China

*Correspondence to*: Zhenghui Xie (zxie@lasg.iap.ac.cn)

**Abstract.** A scheme describing the process of stream-aquifer interaction was combined with the land model CLM4.5 to investigate the effects of stream water conveyance over riparian banks on ecological and hydrological processes. Two groups of simulations for five typical river cross-sections in the middle reaches of the arid zone Heihe River Basin were conducted.

The comparisons between the simulated results and the measurements from water wells, fluxnet station and remote sensing data showed good performance of the coupled model. The simulated riparian groundwater table at a propagation distance of less than 1 km followed the intra-annual fluctuation of the river water level, and the correlation was excellent ($R^2 = 0.9$) between the river water level and the groundwater table at the distance 60 m from the river. The correlation rapidly decreased as distance increased. In response to the variability of the water table, soil moisture at deep layers also followed

the variation of river water level all year, while soil moisture at the surface layer was more sensitive to the river water level in the drought season than in the wet season. With increased soil moisture, the average gross primary productivity and respiration of riparian vegetation within 300 m from the river at a typical section of the river increased by approximately 0.03 mg C m$^{-2}$ s$^{-1}$ and 0.02 mg C m$^{-2}$ s$^{-1}$, respectively, in the growing season. Consequently, the net ecosystem exchange increased by approximately 0.01 mg C m$^{-2}$ s$^{-1}$, and the evapotranspiration increased by approximately 3 mm d$^{-1}$. Furthermore,

the length of the growing season of riparian vegetation also increased by 2–3 months due to the sustaining water recharge from the river. Overall, the stream-aquifer water interaction plays an essential role in the controlling of riparian hydrological and ecological processes.

## 1 Introduction

Water is indispensable for eco-hydrological system (Milly et al. 2005; Ouyang et al. 2003; Shen and Chen 2010; Zhao and
Cheng 2002). Among variety kinds of water resources, aquifer water and stream water, which constitute more than 30% of the freshwater storage, are key factors in hydrological cycle (Chen and Xie 2010; Schär et al. 1999; Xie et al. 2014; Yu et al.

2014). The aquifer water usually acts as a water buffer reservoir to the ecological and hydrological system (Fan 2015; Tsur and Graham-Tomasi 1991). In humid season, aquifer water can store the excess rainfall, and in arid season, it reversely recharges the wet root-zone soil and sustains the ecosystem above by upwards capillary flux (Nepstad et al. 1994). The stream is also very important in the eco-hydrological system. It continuously transports water from humid region to arid region and supports the ecosystem in the lower-reach area (Contreras et al. 2011; Jobbagy et al. 2011).

The relationship between water in streams and aquifers are close and both resources have important roles in the carbon-water cycle and in supplying human needs (Chen and Xie, 2010, 2012; Yu et al., 2014; Zou et al., 2014, 2015; Xie et al., 2014). In a wet region, rainfall or melting snow can raise the groundwater table to an elevation higher than that of the vicinal stream level, and groundwater can sustain base flow in streams and rivers (Arnold et al., 2000). In an arid region, groundwater is recharged laterally from rivers to unconfined aquifers by the stream water conveyance, which sustains the terrestrial ecosystem along the natural channel (Scanlon et al., 2002; Chen et al., 2004, 2010) and induces an increase of riparian soil moisture, soil evaporation, and vegetation transpiration. The growth of riparian vegetation and subsequently changes in carbon cycle processes respond to the water supplement of streams. Understanding and quantifying the effects of stream water conveyance over riparian banks on ecological-hydrological processes is of significance for water resources management (Baskaran et al., 2009).

To investigate the interaction between groundwater and climate, Liang et al. (2003) and Liang and Xie (2003) presented a new parameterization to represent surface and groundwater dynamics and implemented it into the variable infiltration capacity model. Studies have documented that the interaction between surface water and groundwater significantly affect the partition of the water budget and then the land-atmosphere interaction (Maxwell et al., 2007; Maxwell and Kollet, 2008; Fan and Miguez-Macho, 2010, 2011; Fan et al., 2015). To predict the water table elevation near a river channel in an arid region from river discharge, Xie and Yuan (2010) developed a statistical-dynamical approach, whereas Di et al. (2011) and Xie et al. (2012) each developed a quasi two-dimension and quasi three-dimension variably saturated groundwater flow model. These works focused on the temporal and spatial variation of the groundwater table and soil moisture in a riverbank. However, the impacts of river-aquifer water exchange on ecological-hydrological processes, including energy and vapor fluxes, gross primary productivity (GPP) and net ecosystem exchange (NEE) for the riparian ecosystem are not fully represented in previous research. In this study, a scheme for stream-aquifer water interaction were combined with the Community Land Model Version 4.5 (CLM4.5), which contains descriptions about the energy, biophysical and biochemical processes of the land surface and sub-surface, to investigate the effects of stream-aquifer interaction over 5 cross-sections in the middle reaches of Heihe River Basin, a typical region having an arid climate. Overall, the objectives of the study is: (1) Combining the scheme of stream-aquifer interaction and the land model CLM4.5; (2) Quantifying the magnitudes of the responses of the riparian hydrological and ecological processes to the stream-aquifer water interaction; (3) Quantifying the maximum distance that the stream water lateral flow can affect along the riverbank, and studying the relationship between the magnitude of the effects and the distance to river.

In Sect. 2 of this paper, the model description about the stream-aquifer interaction scheme and CLM4.5 are specifically

described, while some background information about the study domain and the experimental design are described in Sect. 3. Section 4 contains the results of simulations and the corresponding analysis. The conclusions and discussion are presented in Sect. 5.

## 2 Model Description

### 2.1 Community Land Model4.5

The land surface model CLM4.5 was developed by the National Center for Atmospheric Research (Oleson et al., 2013), and is the land component of the Community Earth System Model 1.2.0 (Gent et al., 2011; Hurrell et al., 2013). The CLM4.5 model simulates the biogeophysical exchange of radiation, sensible and latent heat flux; momentum between the land and atmosphere as modified by vegetation and soil; heat transfer in soil and snow; and the hydrologic cycle including precipitation interception, infiltration, runoff, soil water, groundwater table depth and snow dynamics (Lindsay et al., 2014). Bio-geochemical cycles including processes of the carbon and nitrogen cycles, photosynthesis, vegetation phenology, decomposition, and fire disturbances are also presented in CLM4.5. Evapotranspiration simulated by CLM4.5 is partitioned into evaporation and transpiration regulated by stoma physiology and photosynthesis. Specifically, in the CLM4.5 a non-linear groundwater reservoir model is used, but it is basically a one-dimensional model which only explicitly accounts the vertical recharge from soil layer to aquifer on the dynamics of groundwater table. Though a subsurface runoff scheme is applied, it does not explicitly solve the lateral flow and is only suitable in the large-scale modeling. More information about CLM4.5 is contained in the Journal of Climate (http://journals.ametsoc.org /page/CCSM4/CESM1).

### 2.2 Configuration of CLM4.5 for simulation over riverbank

Generally CLM4.5 is used for large-scale simulations (global/continental) using relatively coarse grid resolution (about 0.1-1 degree), and these simulations usually make use of a horizontal-2D grid structure. However, in the investigation of the effects of stream-aquifer water interaction over riverbank (especially the intensities of these effects with different distances to river), only the one-dimensional direction perpendicular to the river is matter. Furthermore, the spatial scale of the stream-aquifer interaction is usually restricted within several hundred meters. So some special modifications and configurations should be conducted to make the model suitable for the one-dimensional and fine-scale simulation.

As an example, to a certain cross-section, we first made the one-dimensional surface dataset used in CLM4.5 simulation for the riverbank using surfacedata-generated tool (Kluzek 2012). The schematic diagram for these one-dimensional grids of the surface dataset was shown in Figure 1c. At this time, the longitude and latitude values of each grid were set arbitrarily because they would be modified later. Then we changed the longitude and latitude of each grid to make them represent the real location of each site over the riverbank. Next, we replaced each grid's the elevation, terrain slope, maximum fractional saturated area, land cover type (bare ground, vegetation, lakes, etc.) and soil type (percentage of clay, silt, sand and soil organic matter) of the surface dataset with high-resolution ASTER Dem Dataset (Hirano et al. 2003; Li et al. 2011),

MICLCover (Ran et al. 2012), HiWATER Land Cover Map (Li et al. 2013), and China Soil Characteristics Dataset (Shangguan et al. 2012). At last, the subsurface runoff scheme in CLM4.5 was turned off because it was not suitable in the fine-scale modeling and would be replaced by the groundwater lateral flow in stream-aquifer interaction scheme (described in the Sect. 2.3, which was the explicit representation of the subsurface process). All the vertical biogeophysical and biogeochemical processes of CLM4.5 was retained because they were not scale-dependent and could be used in any resolution if the corresponding surface dataset was set properly.

## 2.3 Scheme for stream-aquifer interaction and its implementation into CLM4.5

The stream-aquifer water interaction scheme (including groundwater lateral flow) developed by Di et al. (2011) was combined with CLM4.5 (the combined model was called CLM_RIV). We first describe the new model briefly as follows. Based on Darcy's law and the Dupuit approximation (Bear, 1972), the lateral flow between a river and the neighboring groundwater can be expressed as:

$$R(x,t) = \frac{\partial Q}{\partial x} = \frac{\partial}{\partial x}\left(T(x,t)\frac{\partial h(x,t)}{\partial x}\right), x > 0, t \geq 0, \tag{1}$$

while the corresponding initial and boundary conditions are expressed as:

$$h(x,0) = h_0(x), \tag{2}$$

$$h(0,t) = h_{river}(t), \tag{3}$$

where $x$ (L) is the perpendicular distance from the point on a bank to the river channel, $t$ (T) is time, $R(x,t)$ (L/T) is the lateral groundwater recharge (or discharge) rate at point $x$ and time $t$, $Q$ [$L^2$/T] is the lateral flow discharge, $T(x,t)$ ($L^2$/T) is the lateral flow transmissivity, $h(x,t)$ (L) is the groundwater table elevation, $h_0(x)$ (L) is the initial groundwater table elevation and $h_{river}(t)$ (L) is the river water level, as shown in Figures 1a and 1b. If the river water level is higher in elevation than its neighboring groundwater table (as shown in Figure 1a), $R(x,t)$ is greater than zero and the local aquifer is recharged by the stream; otherwise, as shown in Figure 1b, $R(x,t)$ is less than zero and the local aquifer discharges to the stream.

To combine the stream-aquifer interaction scheme with CLM4.5, the continuity Eq. (1) should be discretized over the one-dimensional grids of the surface dataset of CLM4.5 (Figure 1c). Applying the zero-flux boundary condition to the outermost grid of the simulation domain, the discrete formation of Eq. (1) can be written as:

$$\begin{cases} R_{1,n} = \dfrac{\dfrac{T_{0,n}+T_{1,n}}{2} \times \dfrac{h_{rn}-h_{1,n}}{\Delta x/2} - \dfrac{T_{1,n}+T_{2,n}}{2} \times \dfrac{h_{1,n}-h_{2,n}}{\Delta x}}{\Delta x} \\[4mm] R_{i,n} = \dfrac{\dfrac{T_{i-1,n}+T_{i,n}}{2} \times \dfrac{h_{i-1,n}-h_{i,n}}{\Delta x} - \dfrac{T_{i,n}+T_{i+1,n}}{2} \times \dfrac{h_{i,n}-h_{i+1,n}}{\Delta x}}{\Delta x}, 2 \le i \le m-1 \\[4mm] R_{m,n} = \dfrac{\dfrac{T_{m-1,n}+T_{m,n}}{2} \times \dfrac{h_{m-1,n}-h_{m,n}}{\Delta x}}{\Delta x} \end{cases}, \quad (4)$$

where $i$ is the number of the grid that is successively added with the increasing distance from grid to channel (Figure 1c), $m$ is the farthest grid from the river channel in the model (i.e., the outermost grid of the simulation domain), $n$ is the number of the time step, $R_{i,n}$ (L/T) is the lateral groundwater recharge (or discharge) rate of grid $i$ at the nth time step, $T_{i,n}$ (L²/T) is the lateral flow transmissivity, $h_{i,n}$ (L) is the groundwater table elevation, $h_{rn}$ (L) is the river water level (which is another boundary condition of the simulation and will be discussed in Sect. 3.2), and $\Delta x$ (L) is the side length of each model grid.

The variables $h_{i,n}$, $T_{i,n}$ and $R_{i,n}$ ($i > 0$) in Eq. (4) are linked to CLM4.5 as follows. The water table elevation $h_{i,n}$ is easily obtained by subtracting the groundwater table depth from the ground elevation as:

$$h = h_e - z_{wt}, \quad (5)$$

where $h_e$ (L) and $z_{wt}$ (L) are, respectively, the ground elevation and current groundwater table depth of the grid calculated by CLM4.5. To obtain the lateral flow transmissivity $T_{i,n}$, we considered two cases in the model. In case A, the groundwater table is within the soil layers of the model (i.e., water table lies within 3.8m from surface) and the transmissivity can be expressed as:

$$T = T_1 + T_2, \quad (6)$$

$$T_1 = \begin{cases} K_i \times \left(z_{h,i} - z_{wt}\right) + \sum_{j=i+1}^{10} K_j \Delta z_j, i < 10 \\[3mm] K_{10} \times \left(z_{h,10} - z_{wt}\right), i = 10 \end{cases}, \quad (7)$$

$$T_2 = \int_0^\infty K(z')\, \mathrm{d}\,z' = \int_0^\infty K_{10} e^{-\frac{z'}{f}}\, \mathrm{d}\,z' = K_{10} f, \quad (8)$$

where $T_1$ (L²/T) and $T_2$ (L²/T) are respectively the lateral flow transmissivity within and outside the 10th soil layers of CLM4.5, $j$ is the number of soil layer denoted by CLM4.5, $K_j$ (L/T) and $f$ (L) are, respectively, the lateral hydraulic conductivity of the $j^{th}$ soil layer and the e-folding length (which will be discussed later), $\Delta z_j$ (L) is the thickness of the $j^{th}$

soil layer, $i$ is the soil layer where the groundwater table lies, $z_{h,i}$ (L) is the lower boundary depth of the $i^{th}$ soil layer, $z'$ (L) is the relative depth to the bottom boundary of the $10^{th}$ soil layer (where $z' = z - 3.8$, $z > 3.8$ m), and $K(z')$ (L/T) is the lateral hydraulic conductivity at relative depth $z'$. Based on Fan et al. (2007), we also applied an estimation of the lateral hydraulic conductivity at depth below the $10^{th}$ soil layer in Eq. (8) as:

$$K(z') = K_{10}e^{-\frac{z'}{f}},$$  (9)

In CLM4.5, only the vertical hydraulic conductivity is provided. So to obtain the lateral hydraulic conductivity $K_j$ of each soil layer, we applied the assumption of Fan et al. (2007) such that the lateral conductivity is related to the vertical hydraulic conductivity and the content of clay for local soil as:

$$K_j = K_j' \times P_{clay},$$  (10)

where $K_j'$ (L/T) is the vertical hydraulic conductivity provided by CLM4.5 and $P_{clay}$ is the percentage of clay in local soil, as provided by surface data of CLM4.5. The e-folding length $f$ in Eq. (8) is a parameter representing the local sediment-bedrock profile, which is complex depending on tectonics, weathering and erosion-deposition processes. In this study, we simply implemented an estimation of Fan et al. (2007) to relate e-folding length to terrain slope as:

$$f = \begin{cases} \dfrac{20}{1+125\beta}, \beta \le 0.16 \\ 1, \beta > 0.16 \end{cases},$$  (11)

where $\beta$ (radian) represents the terrain slope and can be obtained from the surface data of CLM4.5.

In case B, where the groundwater table is positioned below the 10th soil layer of CLM4.5, the $T_{i,n}$ can be calculated as:

$$T = \int_{z_{wt}-3.8}^{\infty} K(z')\,\mathrm{d}z' = \int_{z_{wt}-3.8}^{\infty} K_{10}e^{-\frac{z'}{f}}\,\mathrm{d}z' = K_{10}fe^{\frac{3.8-z_{wt}}{f}}.$$  (12)

We also applied the parameterization of Eq. (9) in Eq. (12).

In Eq. (4), $T_{0,n}$ (L²/T) is the flow transmissivity of the river with respect to groundwater-river exchange. Based on Xie and Yuan (2010), flow transmissivity can be expressed as:

$$T_0 = K_r w,$$  (13)

where $w$ (L) is the river width obtained from measured data and $K_r$ (L²/T) is the hydraulic conductivity at the river bed (which will be discussed in Sect. 3.2).

Finally, the lateral water recharge (or discharge) rate $R_{i,n}$ in Eq. (4) is linked to CLM4.5 as follows:

$$\begin{cases} z_{wt\_new} = z_{wt\_ori} - \dfrac{R \times \Delta t}{s_y} \\ W_{new} = W_{ori} + R \times \Delta t \end{cases},$$ (14)

where $\Delta t$ (T) is the time step of CLM4.5, $s_y$ is the aquifer specific yield calculated by CLM4.5, $z_{wt\_ori}$ (L) and $z_{wt\_new}$ (L) are, respectively, the original simulated groundwater table depth by CLM4.5 and the updated value after considering the later flow flux, and $W_{ori}$ (L) and $W_{new}$ (L) are, respectively, the original simulated aquifer water storage by CLM4.5 and the

5 updated value after considering the lateral flow flux.

Equations (4) to (14) are applied in CLM4.5 to renew the values of groundwater table depth and aquifer water storage at every time step. Other hydrological and ecological variables will be in turn be modified by these changes as the model continues to operate.

Besides the hydrological and ecological processes, the thermal processes of soil and ground are also affected by the

10 stream-aquifer water interaction. In CLM4.5, the ground and soil heat transfer algorithm is applied on the vertical direction as:

$$c\frac{\partial T}{\partial t} = \frac{\partial}{\partial z}\left[\lambda \frac{\partial T}{\partial z}\right],$$ (15)

where $z$ (L) is in the vertical direction and is positive downward, $T$ (K) is the temperature, $c$ (J L$^{-3}$ K$^{-1}$) is the volumetric soil heat capacity, $\lambda$ (W L$^{-1}$ K$^{-1}$) is the thermal conductivity and $t$ (T) is time. The upper (surface) boundary condition of Eq. (15)

is got from radiation calculation of CLM4.5, and the lower boundary condition is set as zero-flux situation. Both the thermal properties of $c$ and $\lambda$ depend on the soil water content as follow (assuming no soil ice for concise expression):

$$c = c_s\left(1 - \theta_{sat}\right) + c_{liq}\theta_{liq},$$ (16)

$$\lambda = K_e\lambda_{sat} + \left(1 - K_e\right)\lambda_{dry},$$ (17)

$$K_e = \lg\left(\frac{\theta_{liq}}{\theta_{sat}}\right) + 1,$$ (18)

where $c_s$ (J L$^{-3}$ K$^{-1}$) and $c_{liq}$ (J L$^{-3}$ K$^{-1}$) are respectively the heat capacity of soil solids and liquid water, $\theta_{sat}$ (L$^3$ L$^{-3}$) and $\theta_{liq}$ (L$^3$ L$^{-3}$) are respectively the saturated soil moisture and current soil liquid water content, $\lambda_{sat}$ (W L$^{-1}$ K$^{-1}$) and $\lambda_{dry}$ (W L$^{-1}$ K$^{-1}$) are respectively the saturated thermal conductivity and dry thermal conductivity, and $K_e$ is the Kersten number. More detailed information about the heat transfer calculation can be found in the Chapter 6 of technical description of CLM4.5 (Oleson et al. 2013). As shown by equations (15)-(18), soil moisture $\theta_{liq}$ impacted by stream-aquifer water interaction would

indirectly affect the simulated temperature and the other thermodynamic variables. Currently, the river temperature and the horizontal heat transfer are not included, but will be incorporated to our model in the future.

## 3 Study domain and experimental design

### 3.1 Study domain

The Heihe River Basin is the second largest inland river basin in an arid area in Northern China. It is located between 96°42′E and 102°00′E and between 37°41′N and 42°42′N (Lu et al., 2003) (Figure 2). The basin covers 116,000 km$^2$ and lies to the east of the Shule River Basin and west of the Shiyan River Basin (Chen et al., 2005). In the upper reaches of the basin with obvious vertical zonal divisions, the mean annual precipitation is approximately 200 mm at elevations from 2000 m to 3200 m, and about 500 mm at elevations between 3200 m and 5500 m. The upper reaches are the main water resource of the entire basin (Wu et al., 2010). In the middle reaches, the elevation decreases from 2000 m to 1000 m and the precipitation correspondingly decreases from 200 mm to less than 100 mm in the direction from south to north (Li et al., 2001). The lower reaches, whose mean altitude is approximately 1000 m, is an arid region with a mean annual precipitation of only 42 mm according to statistics from meteorological stations (Qi and Luo, 2005).

In this study, five typical river cross-sections were chosen as test sites to simulate using our CLM_RIV model. These sites were named, respectively, 213 Bridge, 312 Bridge, Tielu Bridge, Pingchuan Bridge and Gaotai Bridge, and all are located on the middle reaches of the Heihe River Basin. Among these sites, the 213 Bridge section was chosen to test the model's sensitivity, but all the five cross-sections were used in the actual model runs. The locations of these sections and relevant information about them are shown in Figure 3 and Table 1, respectively.

### 3.2 Experimental design

Some ideal experiments to test the model sensitivity to river water level and river bed water conductivity were established for the 213 Bridge section. The CLM_RIV model was run at this section to simulate a riparian zone within 3000 m of the southeast riverbank using a horizontal resolution of 60 m. The simulation period covered the whole year of 2012 using a time step of 1800 s. The atmospheric forcing data were obtained from the China Meteorological Administration Land Data Assimilation System (CLDAS) and developed by the National Meteorological Information Center (NMIC). This high-quality data set combines field observations, remote sensing data and numerical products at a horizontal resolution of 0.0625 degrees. Initial conditions for the simulation were obtained from a 700-year "spin-up" run conducted using the original version of CLM4.5 (without groundwater lateral flow) and cyclically using the CLDAS dataset. The choice of 700 "spin-up" years was based on the user's guide of CLM (Chapter 4 of Kluzek 2013) showing that when the biogeochemistry carbon-nitrogen module of CLM is turned on, the model should be at least run for 700 years to get a steady state because the magnitudes of carbon and nitrogen fluxes are very small (Oleson et al. 2013). For each situation—case (**a**): the river recharging groundwater and case (**b**): the groundwater recharging river—we conducted two sensitivity experiments for each case. The first of these examined the sensitivity of the model predictions to changes in the river water level. Four constant river elevations were considered: in the case (**a**), the four river elevations were $h_r$ = 1493.1 m, 1492.1 m, 1491.1 m and 1490.1 m, and in the case (**b**), the four river elevations were $h_r$ = 1483.1 m, 1478.1 m, 1473.1 m and 1468.1 m. The hydraulic

conductivity of the river bed ($K_r$) was fixed at 7.4 m d$^{-1}$ for both cases. The second experiment tested the sensitivity of the model to changes in the hydraulic conductivity of the river bed. In this experiment, the boundary condition of the river water level was fixed at $h_r$ = 1491.1 m in the case (**a**) and $h_r$ = 1478.1 m in the case (**b**). The four sets of river hydraulic conductivities were prescribed: $K_r$ = 3 m d$^{-1}$, 6 m d$^{-1}$, 12 m d$^{-1}$ and 24 m d$^{-1}$ both for cases.

Then to investigate the eco-hydrological effects of stream-aquifer interaction, a "realistic" simulation and a "control" simulation using CLM_RIV were conducted. The realistic simulation (called TEST) reproduced processes of stream-aquifer interaction and groundwater lateral flow; the control simulation (called CTL) did not take the stream-aquifer interaction into consideration (assuming no water flux between stream and riverbank as boundary condition) but also accounted the groundwater lateral flow over riverbank. For the river grid cells in the middle of each section, no simulations with CLM4.5

were performed. In the Each simulation covered a period of a whole hydrological year from 1 July 2012 to 30 June 2013 using a time step of 1800 s. The models were run at the five sections to simulate both sides of the river within a distance of 3000 m from the river channel using a horizontal resolution of 60 m. As with the sensitivity tests, atmospheric forcing data were used from CLDAS as developed by NMIC. However, instead of using the default surface dataset of CLM4.5 (Oleson et al., 2013), we replaced the data of elevation, terrain slope maximum and fractional saturated area with ASTER Dem Dataset

(30-m resolution, Hirano et al. 2003; Li et al. 2011), land cover data with MICLCover (1-km resolution, Ran et al. 2012) plus HiWATER Land Cover Map (30-m resolution, Li et al. 2013) and soil data with China Soil Characteristics Dataset (1-km resolution, Shangguan et al. 2012). The soil types and vegetation types over both sides of the selected sections are shown in Table 2. Other model parameters, such as the parameters related to atmospheric boundary layer, hydrology, thermodynamics and vegetation (including root length density), were set as the default settings of CLM4.5. Detailed information about these

parameters could be found in the technical description of CLM4.5 (Oleson et al. 2013).

    In the TEST and CTL simulations using CLM_RIV, the lateral hydraulic conductivity of river bed ($K_r$) was set to 7.4 m d$^{-1}$ based on research of Xie and Yuan (2010). The boundary conditions of river water levels ($h_r$) for the five sections were obtained from the data set of the hydrometeorological observation network, which is operated by Heihe Watershed Allied Telemetry Experimental Research (HiWATER, Li et al., 2013; Liu et al., 2014). The observations covered all time periods of

our simulations with a time interval of 1800 s. First, the TEST and CTL were started from the default initial condition of CLM4.5 (seen in Oleson et al. 2013) and run 700 years under each configuration (with and without stream-aquifer water interaction), cyclically using the atmospheric forcing and observed water level data. Then, the TEST and CTL would start their formal runs from 1 July 2012 to 30 June 2013 using the restart files produced by the former 700-year spin-up.

## 4 Results

**4.1 Validation**

First, we validated our model using results from the sensitivity experiments. Both the responses of groundwater table in short-term (7 days) and long-term (160 days) simulations were displayed. The results from case (**a**) (river recharging

groundwater) were plotted in the Figure 4. Figure 4a-4h show the time series of the simulated groundwater table depths for each grid cell in the first sensitivity experiment ($h_r$ was varied and $K_r$ was held as constant). From the figures, groundwater table depth near the river channel is significantly reduced (groundwater head is elevated) as the river water level increases. This is because, as Eq. (1) shows, the higher river water level induces a greater hydraulic gradient, which enhances lateral recharge to the riparian aquifer. This effect is significant in both short-term and long-term simulations indicating the essential role of river level in the controlling of riparian water table. Figure 4i-4p show the time series of the simulated groundwater table depths for each grid cell in the second experiment ($h_r$ was held as constant and $K_r$ was varied). From the figures, the effect of $K_r$ is significant over the short-term simulation (Figure 4i-4l): As the $K_r$ ranged from 3 m d$^{-1}$ to 24 m d$^{-1}$, the time spent by the nearest grid (to river) to get the equilibrium state is shortened from 2 days to 0.5 days. However, after long-term simulation (Figure 4m-4p), the groundwater table depths are similar for all values of $K_r$ indicating that equilibrium state of groundwater table along the river channel is not very sensitive to $K_r$ compared with $h_r$. This is because, river bed water conductivity $K_r$ only connects the river and the nearest model grid to the river, while the rest of grids (not next to river) are not directly influenced by $K_r$ and more affected by the lateral hydraulic conductivity $K$ of the riverbank soil (in Eq. (9)). The results from case (**b**) (groundwater recharging river) were plotted in the Figure 5. The conclusions from Figure 5 are similar as Figure 4: River level is matter over both short-term and long-term simulations in the controlling of riparian water table, while the river bed water conductivity is more important in the controlling of short-term water table variation than the controlling of long-term water table equilibrium. Figure 4 and Figure 5 jointly validated that our model could reasonably reproduce both the processes of river recharging groundwater and groundwater recharging river.

Next, we tested our results from the realistic simulation (TEST) using observed data. First of all, we used observation data from the eddy covariance (EC) and automatic weather station (AWS) system of the Bajitan Gobi Desert station (Liu et al., 2011; Li et al., 2013), a part of hydrometeorological observation network operated by HiWATER, to validate our simulation. The Bajitan Gobi Desert station is located at 100.3042 °E, 38.9150 °N (displayed in Figure 3) and an elevation of 1562 m. The station is on the northwest riverbank of the first section (213 Bridge) in our simulation at a distance of approximately 2800 m from the channel. The station contains a 10-m flux tower equipped with a series of EC instruments for flux measurements, and meteorological instruments for regular weather measurements as well as soil temperature and moisture. The underlying surface of this site is Gobi Desert soil with scarce grass and there are few human activities nearby, which benefitted our validation because anthropogenic effects are not considered in the simulation. Figure 6 shows the daily variations in the observations of surface soil temperature, surface soil moisture, sensible heat flux and latent heat flux at the Bajitan Gobi station against the corresponding simulated values from the CTL and TEST runs. The initial observation times of the EC and AWS system were, respectively, 14 August 2012 and 19 September 2012, and there was a successive period near June 2013 with missing measurements for both sensible and latent heat flux. Figures 6a and 6b show that our model can correctly adjust the surface temperature throughout the year but yields surface soil moisture predictions that have a significant positive bias in spring and winter. Despite this, TEST can generally capture the peak value of soil moisture induced by rain events. Figure 6c shows that our model is credible for sensible heat flux simulation, albeit with

underestimation of this parameter in winter. Figure 6d shows that TEST also simulates the latent head flux well in the rain season, but gives a negative bias in the arid season. Compared with CTL, simulated results of the sensible and latent heat fluxes from TEST are closer to observations, while the results of surface soil temperature and moisture are not distinguished between CTL and TEST. Overall, the TEST simulation demonstrated a reasonable ability of CLM_RIV to reproduce the observations of important parameters, especially in the wet season when the eco-hydrological effects of stream-aquifer water interaction are dominant.

Next, we tested the ability of our model to simulate the groundwater table, which is a key factor in ecological and hydrological effects. We compared the results from both the TEST and CTL simulations with groundwater head elevation and groundwater table depth data from observation wells distributed over the middle reaches of the Heihe River Basin (Zhou et al., 2013). There were 46 wells within our simulation domain of the five sections. Figure 7a shows the annual values of our simulated groundwater head elevation from both TEST and CTL runs against the observed groundwater heads at the 46 wells. As shown, if the stream-aquifer water transfer is not accounted (as in the CTL run), there is a significant underestimation of water head at nearly all sites. When river-groundwater exchange is considered (as in the TEST simulation), the negative biases are much reduced because the water transfer raises the water table, and the modeled groundwater levels are very close to the observations for most wells. However, there are still a few meters of deviation between TEST simulated levels and observed levels. The conclusions above were also shown, more apparently, by the comparison of groundwater table depth in Figure 7b. Figure 7c shows the spatial distribution of groundwater table depth from observation, TEST and CTL over the Gaotai Bridge, along which the most number of wells were deployed. We can also see that the systematic errors of simulated groundwater tables were obviously reduced along the whole riverbank after the stream-aquifer water interaction was accounted, though a few meters of deviation still existed. The deviation of these results may come from the chosen saturated hydraulic conductivity values, which in this study were chosen a priori and as such not optimized in any kind of manner.

Next, we checked the model's ability to simulate spatial variability by comparing simulated ground temperature from the CTL and TEST runs with high-resolution remote sensing data from the Advanced Spaceborne Thermal Emission and Reflection Radiometer (ASTER) launched by the United States National Aeronautics and Space Administration (Tachikawa et al., 2011). The ASTER data had been post-processed for the Heihe River Basin by Li et al. (2014). Ground temperature measurements at 90-m resolution were available for five satellite transit events during the summer of 2012. We used relative temperature of the nearest grid to the stream to emphasize spatial variability. The northwest (left) riverbank of the 213 Bridge station was chosen for our comparison because human activities could be neglected there. Figure 8 shows that in four of the five events TEST successfully simulated the increase in ground temperature as distance from the channel increases, while the CTL could not reproduce this spatial variability. However, in the fourth event, the spatial variability predicted by the TEST simulation is much lower than that indicated by ASTER data. This may be caused by the fact that ASTER data are not processed with a cloud mask, which causes overestimation of the cooling effects of streamflow on a cloudy day (Li et al., 2014).

## 4.2 Eco-hydrological effects of stream-aquifer water interaction

### 4.2.1 Intra-annual responses to river water level

First, we examined the inter-annual responses of eco-hydrological characteristics to river water level variations. Figure 9 shows the intra-annual variations (at 1800-s intervals) of water heads and water table depth at 30 m, 90 m, 210 m and 450 m from the channel on the left riverbanks of streams at the five stations, as well as the observed river water levels. As Figure 9a-e show, the 30-m water heads are tightly connected with river levels and have slightly lower elevations and change-frequencies. The 90-m water heads also follow the river level fluctuations but with some time lags, and the elevations are much lower than the river levels and more resistant to change. At 210 m and 450 m from the stream, there is no discernable relation between water table heads and river water levels, and the former are very stable within the year. The performances of simulated water table depth in Figure 9f-j are similar as the water head elevations. The Figure 9 means the region that can receive the intra-annual signal of river level changes by stream-aquifer interaction is restricted within a limited distance from the channel, and the response to this signal is stronger closer to the river than farther away. The time correlation coefficients between groundwater tables across the left riverbanks and the river levels of the five sections are plotted in Figure 10. Considering the time lags of the signal transduction, we used the maximum value of cross-correlation coefficients with time lags from 0 to 3 months (at 1800-s intervals). The standard line where the correlation coefficient passes the 95% confidence level of the Student's t test is also plotted in Figure 10. As shown in Figure 10, the correlation coefficients between the groundwater tables and river levels are more than 0.9 for locations very near to streams, but decrease rapidly as distances from channels increase. The left riverbanks of the 213 Bridge and Pingchuan Bridge stations are least impacted by intra-annual river fluctuations; only at locations within 200 m from streams at these stations do correlation coefficients pass the Student's t test. The most affected riverbank is located at Tielu Bridge station, where intra-annual river level fluctuations influence the water table elevations as far as 450 m from the stream. Nonetheless, the area impacted by intra-annual river water level fluctuations (i.e., a zone within 450 m of a stream) is much smaller than that impacted by stream-aquifer exchange (i.e., a zone extending to 1800 m from a stream, see Section 4.2.2.).

We then examined the responses of other eco-hydrological characteristics to intra-annual river water level changes. To highlight the outcomes, we show the simulation results at two rather contrasting stations, Tielu Bridge and 213 Bridge; these stations demonstrated the longest and shortest propagation distances, respectively, for river level fluctuation (Figure 10). We plot the area-averaged data within a 300-m range from both sides of the streams.

Figure 11 shows the time series of selected daily ecological and hydrological variables predicted by TEST and CTL simulations, as well as the river levels and precipitation within the simulation period for the Tielu Bridge section. Figures 11c and 10d show that the effects of stream-aquifer interaction on surface soil water and surface ice, respectively, are dominant in spring, autumn, and winter. As expected, the effects on surface soil ice are especially noticeable in winter, with values predicted by the TEST simulation nearly five times those predicted by CTL. The relative lack of influence of the high river water level of summer (Figure 11a) on soil water seems contradictory, but can be explained by the precipitation variation

shown in Figure 11b; in summer, surface soil is wetted most by precipitation and stream water contributes relatively less to this effect, while in other seasons the stream water can significantly affect the surface soil water (and ice) because rain events are sparse. These conclusions can be checked in Figure 11e, which shows that the effects of stream-aquifer interaction are perennially apparent on deep soil water that is much less affected by rain. Figure 11g shows that ground temperature is cooled by stream water in spring and summer and warmed in winter, though the amplitudes of these changes are slight compared with seasonal temperature variation. The higher specific heat capacity induced by wetter soil makes soil temperatures more resistant to the influence of air temperature change than when the soil is dry.

Intra-annual impacts on GPP and ecosystem respiration (RE) are shown in Figures 11h and 11i, respectively. Generally, GPP and RE are both strengthened by stream-aquifer water interaction all year except in winter, and the increased GPP (approximately 0.03 mg C $m^{-2}$ $s^{-1}$ in the growing season) is higher than RE (approximately 0.02 mg C $m^{-2}$ $s^{-1}$) most of the time. These differences enhance the NEE by approximately 0.01 mg C $m^{-2}$ $s^{-1}$ in the growing season, which means that riparian plants fix more $CO_2$ from May to September than at other times of the year (as Figure 11j shows). However, there is a time period from March to April when RE is enhanced by stream water supplement, while GPP is unaffected. This time lag causes the riparian vegetation to act as a strong carbon source in this period (Figure 11j) instead of a sink as at other times of the year.

The incremental leaf area index (LAI) and evapotranspiration by water recharge from the river are shown in Figures 11k and 10l, respectively. The LAI is much increased from April to December relative to other times and the stream water supplement can even advance the beginning, and delay the ending, of the growing season for 1–2 months (Figure 11k). Predictions from the TEST simulation indicate that LAI is zero near September 2012, corresponding to the dry river water condition around this time (Figure 11a); this result underlines the high sensitivity of riparian plant growth to the stream-aquifer water interaction. Figure 11l shows that evapotranspiration variability within the year is also highly related to the fluctuation in river level, reemphasizing the key functions of environmental flows for an ecological system.

Figure 12 shows the time series of selected daily ecological and hydrological variables predicted by TEST and CTL simulations, as well as the river levels and precipitation within the simulation period for the 213 Bridge section. The conclusions based on TEST and CTL simulations for Tielu Bridge are generally applicable to the section at 213 Bridge as shown in Figure 12, which means that the intra-annual responses of eco-hydrological elements to river water level changes are similar at a wide range of sections in this arid region. However, due to the propagation distance of river level fluctuation at the 213 Bridge section being much shorter than at Tielu Bridge (Figure 10), the strength of these hydrological and ecological responses is significantly weaker at 213 Bridge than at Tielu Bridge. The differences can be observed by comparing Figures 11 and 12.

### 4.2.2 Annual averaged effects of stream-aquifer water interaction along riverbanks

After studying the intra-annual responses of the riparian eco-hydrological system to river water fluctuation, we examined the annual averaged effects of stream-aquifer water interaction on riparian eco-hydrological elements along riverbanks.

Figure 13 shows the differences of annual water head between predictions from TEST and CTL simulations and the terrain elevations along the five sections. All sections show stronger effects of elevated water tables closer to the stream than farther away. The water exchange from stream to aquifer can increase the water head at the grid nearest to the stream (30 m from the channel) by 13 m to 22 m. Furthermore, all cross-sections show water table elevations increased by more than 8 m even at sites nearly 2 km from channels. When averaged for the area within 1800 m from either side of the river channel, the groundwater tables rose by approximately 10–20 m at the five sections. These results show that the effects of stream-aquifer water interaction on annual averaged groundwater levels can spread very far by groundwater lateral flow. Thus groundwater studies must consider the impacts of water exchange between a riverbank and river, a point also stressed by other researchers (Miguez-Macho et al., 2007; Chen et al., 2010; Di et al., 2011). As shown by Figure 13, the relationship between the curve shape of elevation and water head along riverbank can be generally figured out: When the terrain is relatively flat, an apparent curvature of water head is occurred, such as the left side of 213 Bridge and Pingchuan Bridge and the right side of Tielu Bridge; when the curvature of terrain is obvious, the curve of water head is relatively flat such as the left side of 312 Bridge and Tielu Bridge and the right side of Pingchuan Bridge. However, the curve shape of water head is determined by multi-factors such as the groundwater recharge, soil type and aquifer thickness. The topography is only (maybe the most) influential one of them. This topographic factor also explained why the effects of river water conveyance are not symmetrical over the left and right sides.

Figure 14 shows the differences of summer and winter soil moisture (both liquid water and ice are included) predicted by TEST and CTL simulations along the five sections. Predictions at two depths (2 cm and 100 cm) are chosen to represent the surface and deep soil layers, respectively. Figures 14a–14e show that in summer, the deep soil moisture is increased by stream water from 0.08 $m^3$ $m^{-3}$ to 0.16 $m^3$ $m^{-3}$ at the grid closest to the channel, and that this wetting effect is weaker as the distance from the river increases. Averaged for the region within 1 km from the stream, the deep soil is wetted by river water by approximately 0.05 $m^3$ $m^{-3}$ (a 30% increase) at the riverbank. However, the surface soil moisture is nearly unaffected by stream-aquifer interaction because in summer, surface soil moisture is dominated by precipitation and stream water contributes little to the soil moisture changes. This conclusion is verified in Figures 14f–14j. In winter when rain events are sparse, the wetting effects of stream-aquifer interaction on surface soil moisture are apparent at all sections, though the magnitudes are small (only approximately 0.02 $m^3$ $m^{-3}$, a 10% increase) compared with the wetting effects on deep soil. Wetter soil supplies more water for riparian plant growth and subsistence than dry soil, especially in the growing season in an arid region, which stresses the necessity of stream-aquifer water interaction in supporting the riparian environment.

The annual averaged ecological effects of stream-aquifer water interaction were also evaluated. Figure 15 shows differences in predicted GPP, RE (both autotrophic and heterotrophic respiration are included) and NEE resulting from TEST and CTL simulations for the summer period. Because there is no vegetation on the northwest (right) side of the 213 Bridge station, all the values are zero (Figure 15a). Figure 15 shows that GPP and RE increased as the distance to the channel decreased, while NEE increased (with the ecosystem tending to be a carbon sink) by 0.002–0.005 mg C $m^{-2}$ $s^{-1}$ (100–300%). The impacts are evident within a range of approximately 1 km. The strongest effects appeared at Tielu Bridge station with

increases of more than 0.05 mg C $m^{-2}$ $s^{-1}$ for GPP and 0.04 mg C $m^{-2}$ $s^{-1}$ for RE, and a decrease of about 0.01 mg C $m^{-2}$ $s^{-1}$ for NEE at the grid nearest to the stream. The influences of stream-aquifer interaction on GPP are stronger than they are on RE at all sections; this difference explains why the stream effects on NEE are negative (carbon sink) and means that riparian vegetation can absorb more $CO_2$ and grow better when it is closer to the river. These results highlight the maintenance function of stream-aquifer water interaction for a riparian ecosystem, especially in an arid region.

The simulated effects of stream-aquifer interaction on LAI and canopy transpiration (canopy evaporation is also included) in the summer period are provided in Figure 16. Differences in LAI and transpiration predicted by the TEST and CTL simulations show similar spatial patterns at all sections; in close proximity to the river, LAI and transpiration are increased by supplemental water from the stream. The impacted areas are also within approximately 1 km from the channel for most riverbanks. Averaged over the affected area, the transpiration is enhanced by 0.2–1.0 mm $d^{-1}$ (about 100–200%) and LAI is increased by 0.2–1 in summer. The strongest affected section is Tielu Bridge where the LAI and canopy transpiration increased by approximately 5.0 mm $d^{-1}$ and 4 mm $d^{-1}$, respectively, at the closest grid to the stream (Figure 16c); riverbanks of other sections are less impacted. The similar spatial distributions of LAI and transpiration across riverbanks means that in this arid region, transpiration along the river is mainly controlled by LAI, which will benefit from stream water lateral infiltration. This finding again stresses the essential influence of stream-aquifer water interaction in riparian hydrologic and carbon cycles, as well as in maintaining environmental integrity.

Lastly, we show the effects of stream-groundwater exchange on vertical energy and water fluxes along a river. Figure 17 shows the differences in sensible heat (SH) and latent heat (LH) fluxes predicted by the TEST and CTL simulations for summer and winter. Figures 17a–17e show that the effects on SH and LH in summer display opposite trends along the riverbanks: LH becomes stronger closer to the stream while SH becomes weaker. The stronger LH is due to the enhanced evapotranspiration along the river (Figure 16), which also induces weaker SH. However, the SH and LH trends change in winter. Figures 17f–17j show that both SH and LH exhibit small positive changes closer to riverbanks, though the magnitudes are much smaller than they are in summer; this may be induced by the lower river water level in winter (Figure 9). Because SH and LH are key factors influencing the atmosphere above a plant canopy, local weather and climate would also be modified by the effects of stream-aquifer water interaction; this suggests that when studying local climate in areas that include streams, the effects of surface water should not be ignored.

## 5 Conclusions and Discussion

In this study, we combined a scheme of stream-aquifer water interaction with the land model CLM4.5 to investigate the eco-hydrological effects of stream-aquifer water interaction over riverbank. After sensitivity tests for selected parameters demonstrated the reliability of the combined model (CLM_RIV), the model was used to make two simulations to detect the effects of stream-aquifer water interaction on ecological and hydrological processes on riparian banks at five different locations. One simulation was "forced" using observed river water levels. The other "control" simulation did not take

stream-aquifer water exchange into consideration. Both simulations covered a period from July 2012 to June 2013. Comparisons of simulation outputs and observations from EC and AWS systems, water wells and remote sensing data demonstrated that CLM_RIV shows considerable ability to reproduce the natural conditions along riverbanks.

The main conclusions of this study are as follows. (1) A riparian groundwater table responds to the intra-annual variation in river water level, but the response areas are limited to within 200–450 m from the stream channel. The correlation coefficient between the groundwater table and river level can reach 0.9 at the nearest model grid to the river, but rapidly decreases as the distance from the river increases. Surface soil liquid water in the rain season is less impacted by river level variation than is deep soil water, which follows the river level fluctuation all year. (2) Over a typical riverbank section (Tielu Bridge), averaged GPP and respiration of riparian vegetation within 300 m from the stream increased by approximately 0.03 mg C m$^{-2}$ s$^{-1}$ and 0.02 mg C m$^{-2}$ s$^{-1}$, respectively, in the growing season due to increased soil water, resulting in enhanced NEE of approximately 0.01 mg C m$^{-2}$ s$^{-1}$. Evapotranspiration in this zone also increased (by approximately 3 mm d$^{-1}$). Furthermore, the growing season of riparian vegetation is also extended by 2–3 months due to the sustaining water recharge from the stream, and even a short-term decline in river level can negatively impact LAI near the stream during the growing season. (3) All impacted ecological and hydrological characteristics are restricted to an area within approximately 1 km from the channel, and the effects become stronger as distance to the river decreases. These conclusions highlight the functions of stream-aquifer water interaction on sustaining and controlling the riparian ecological system, and indicate the potential benefits of water regulation, such as through artificial stream water conveyance, to maintain stream flow.

However, there are assumptions and limitations of this study that should be noted. Besides the intrinsic uncertainties of CLM and atmospheric forcing (Bonan et al., 2011, 2013; Mao et al., 2012; Wang et al., 2013), the parameters reflecting the land and river conditions in our scheme, such as $K_j$, $K_r$ and f in Eq. (4)–(14), are highly parameterized based on some simple assumptions to facilitate data collection and computation, while the real states of geological structures and sediment-bedrock profiles are so complex that they are almost impossible to describe accurately. However, the sensitivity experiments and comparison of our results with data from multiple sources (Sect. 4.1) prove that these uncertainties do not significantly affect the simulation ability of CLM_RIV. Another restriction on our results is that human activities, such as irrigation that may take place on riverbanks, are not considered in our model. Such activities could cause our results to deviate considerably from the real situation. Arguably, the aim of this study was to emphasize the effects of stream-aquifer water interaction (which is a totally natural process) on riparian eco-hydrological processes. Thus, ignoring anthropogenic disturbances on riverbanks (such as crop cultivation, irrigation and water diversion), which may interfere with the natural influences we simulated, was a reasonable approach in this research.

Some future studies are also needed. To overcome the uncertainties of parameterization, more systematic experiments to test the sensitivity of model parameters should be conducted, and corresponding observations or more sophisticated estimation approaches for key parameters relating to stream-aquifer interaction are needed. Finally a land-river-atmosphere interaction model that can simulate the water and energy exchange between each component is needed for studying the more comprehensive effects of stream water flows.

**Acknowledgements** This study was supported by the National Natural Science Foundation of China (Grants 91125016, 41575096 and 41305066) and by the Chinese Academy of Sciences Strategic Priority Research Program under Grant XDA05110102. We would like to thank Xing Yuan, Xiangjun Tian and Yuanyaun Wang for their assistance with this work
and helpful discussion.

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

| Number of section | Name | Latitude | Longitude | Width (m) | Riverbank elevation (m) | Bottom elevation (m) | Flow direction |
|---|---|---|---|---|---|---|---|
| 1 | 213 Bridge | 38°54'43.55"N | 100°20'41.05"E | 330 | 1493.1 | 1488.8 | Northeast |
| 2 | 312 Bridge | 38°59′51.71″N | 100°24′38.76″E | 70 | 1402 | 1397 | Northeast |
| 3 | Tielu Bridge | 39°2'33.08"N | 100°25'49.42"E | 50 | 1382 | 1379.25 | Northeast |
| 4 | Pingchuan Bridge | 39°20'2.03"N | 100°5'49.63"E | 130 | 1323.8 | 1319 | West |
| 5 | Gaotai Bridge | 39°23'22.93"N | 99°49'37.29"E | 210 | 1295.5 | 1288.5 | West |

**Table 1: The locations and relevant information about the five selected sections used in simulations.**

| Number of section | Name | Soil type (Left) | Soil type (Right) | Vegetation type (Left) | Vegetation type (Right) |
|---|---|---|---|---|---|
| 1 | 213 Bridge | Sand | Silt | Bare ground | Corn |
| 2 | 312 Bridge | Silt | Silt | Corn | Grass |
| 3 | Tielu Bridge | Silt | Silt | Corn | Grass and corn |
| 4 | Pingchuan Bridge | Silt | Silt | Grass and Corn | Grass and corn |
| 5 | Gaotai Bridge | Sand | Sand | Grass | Corn |

**Table 2: The soil types and vegetation types over both sides of the five selected sections used in simulations.**

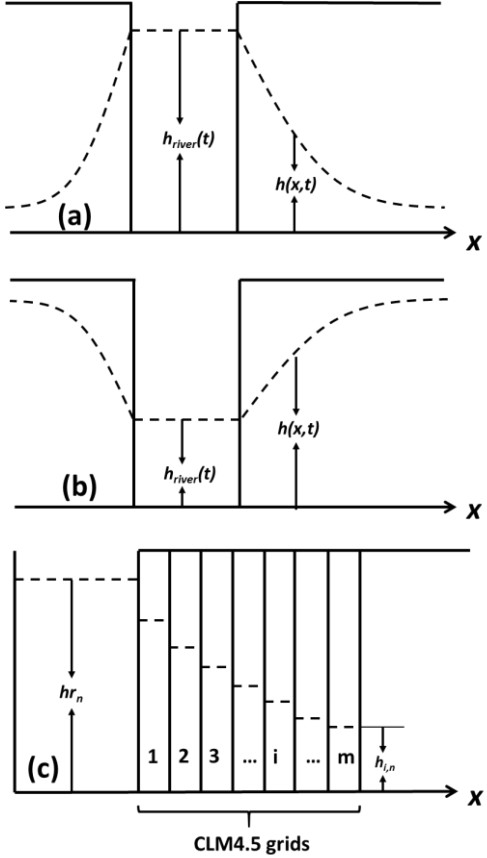

**Figure 1: Schematic representation of stream-aquifer water interaction when (a) the river water level is higher than its neighboring groundwater table and (b) the river water level is lower than its neighboring groundwater table. (c) Schematic diagram for horizontal discrete grid cells of a riverbank. The dash lines represent the water heads.**

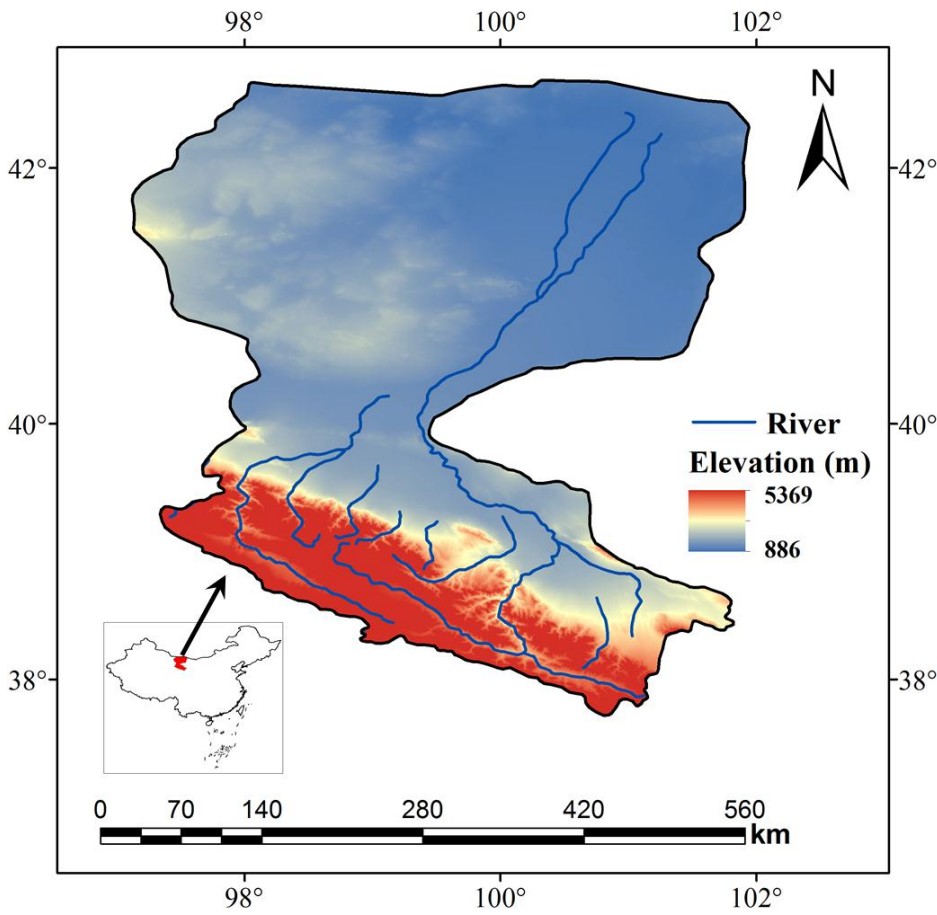

**Figure 2: Study area and location of the Heihe River Basin in northwest China.**

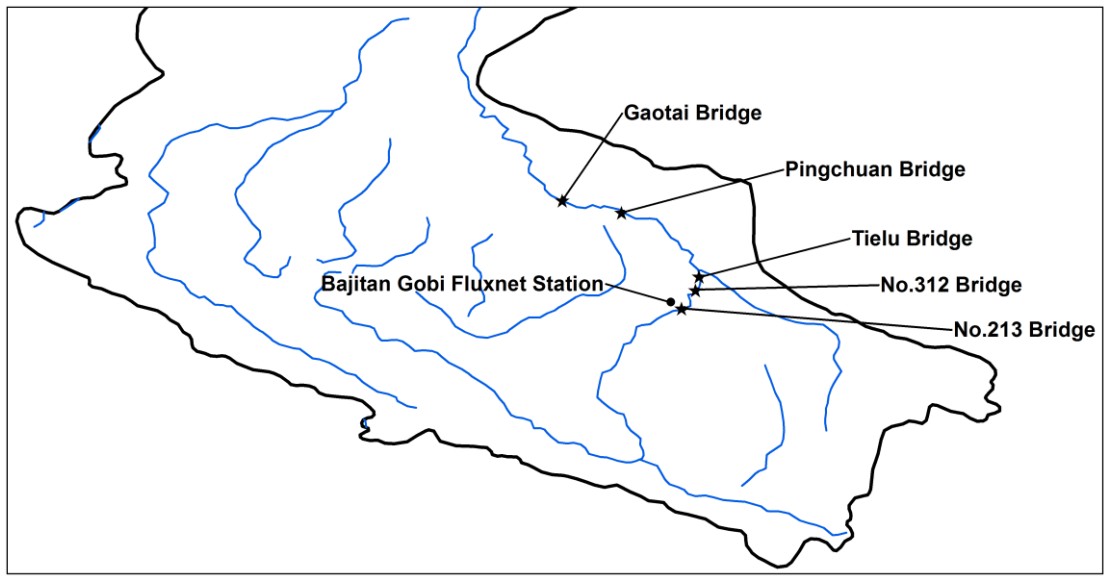

**Figure 3: Locations of the five sections in the middle reaches of the Heihe River that were used for simulations and the location of Bajitan Gobi Fluxnet Station that was used for validation.**

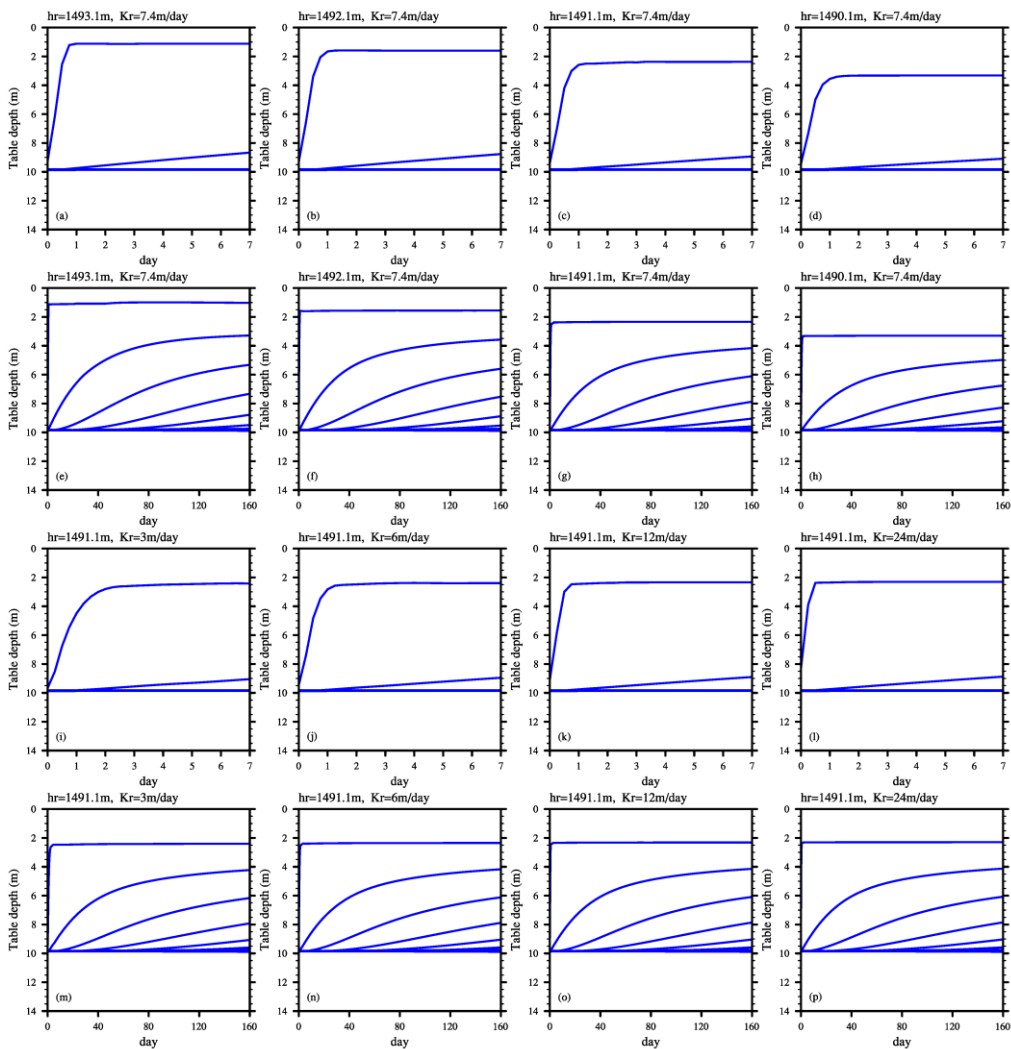

**Figure 4:** (a-d, i-l) Short-term and (e-h, m-p) long-term responses of riparian groundwater table to the river water level hr and river bed hydraulic conductivity Kr in the case of river recharging groundwater. (a-h) Time series of the simulated groundwater table depths for each grid cell in the first sensitivity experiment. (i-p) Time series for the second sensitivity experiment.

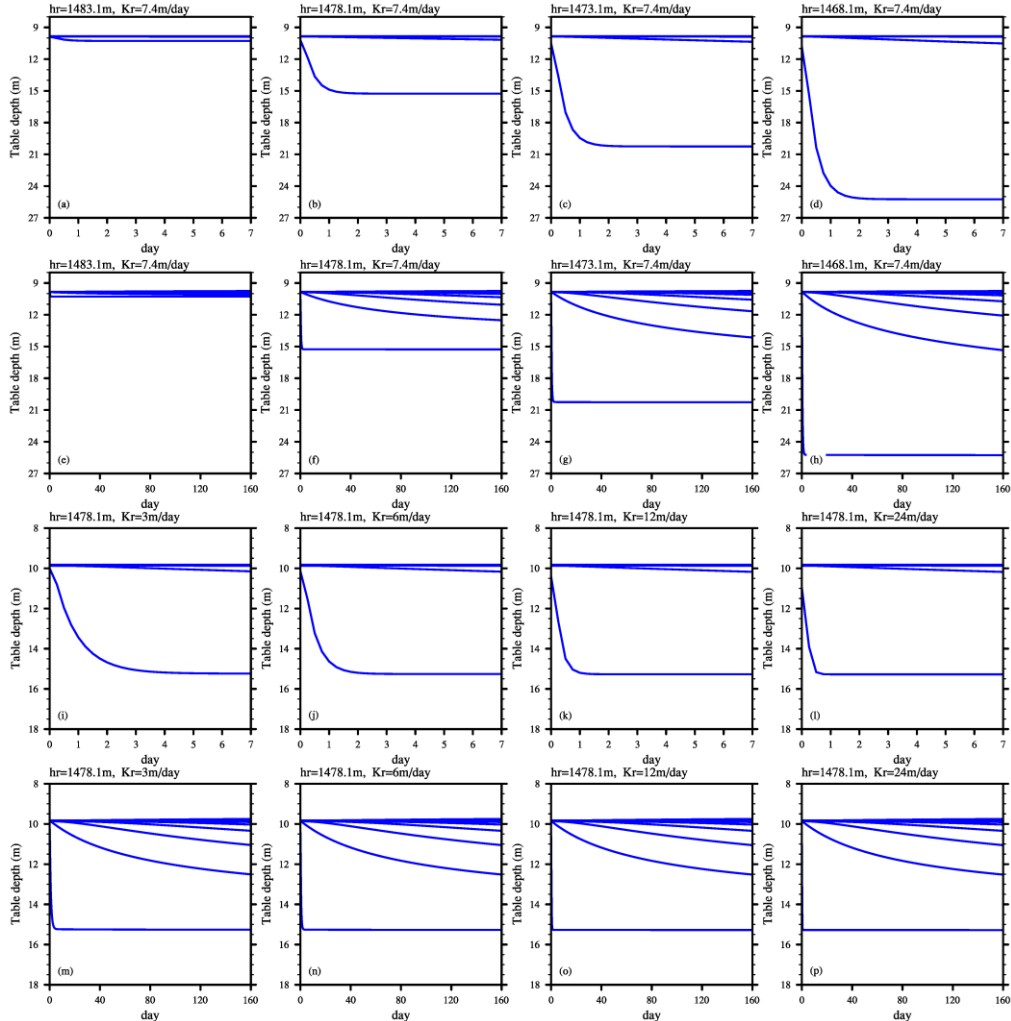

**Figure 5: (a-d, i-l) Short-term and (e-h, m-p) long-term responses of riparian groundwater table to the river water level hr and river bed hydraulic conductivity Kr in the case of groundwater recharge river. (a-h) Time series of the simulated groundwater table depths for each grid cell in the first sensitivity experiment. (i-p) Time series for the second sensitivity experiment.**

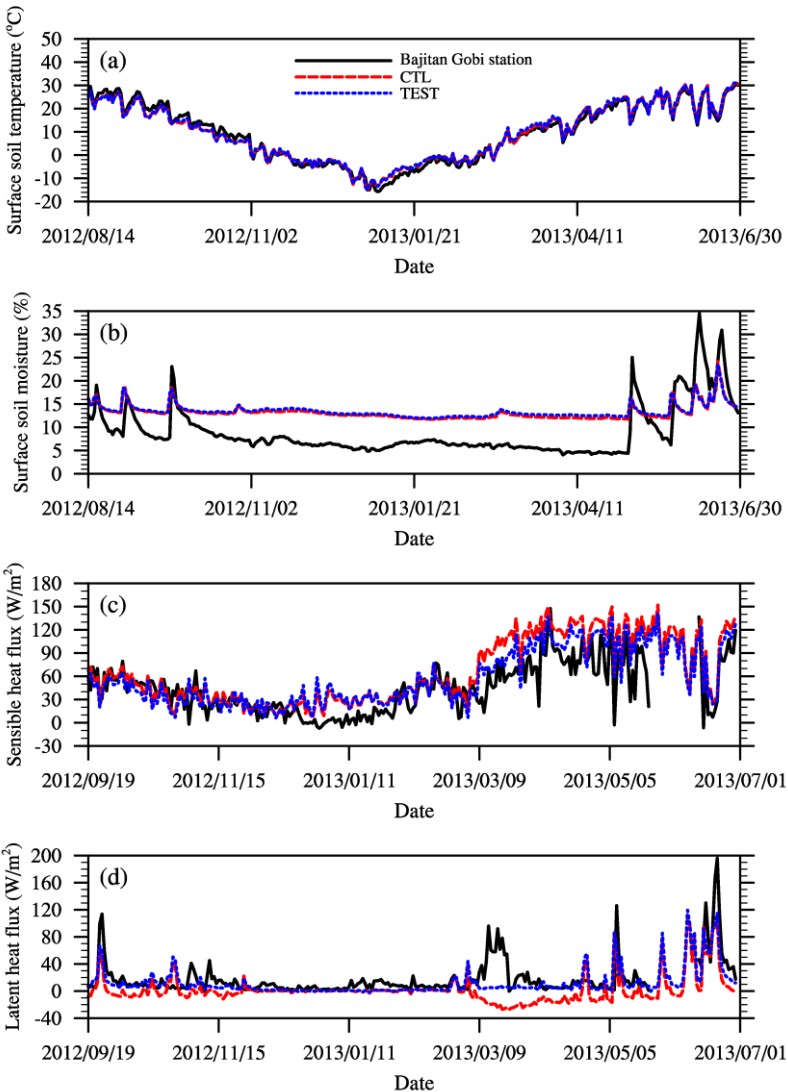

**Figure 6: Time series of the observations from the eddy covariance and automatic weather station systems and results from the CTL and TEST simulations at Bajitan Gobi station for (a) surface soil temperature, (b) surface soil moisture, (c) sensible heat flux and (d) latent heat flux.**

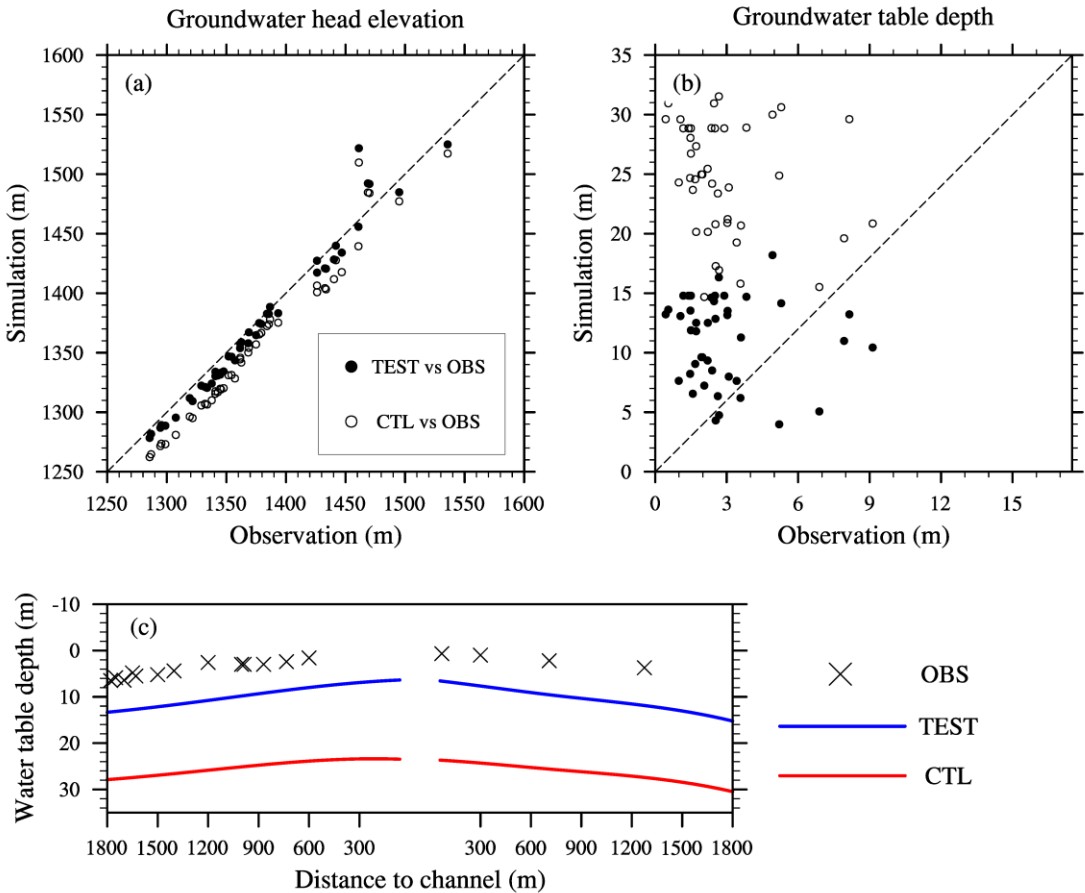

**Figure 7: (a) Annual groundwater head elevation, (b) groundwater table depth predicted by TEST and CTL simulations against observed climatology water head data from 46 observation wells and (c) spatial distribution of groundwater table depth from observation, TEST and CTL over the Gaotai Bridge.**

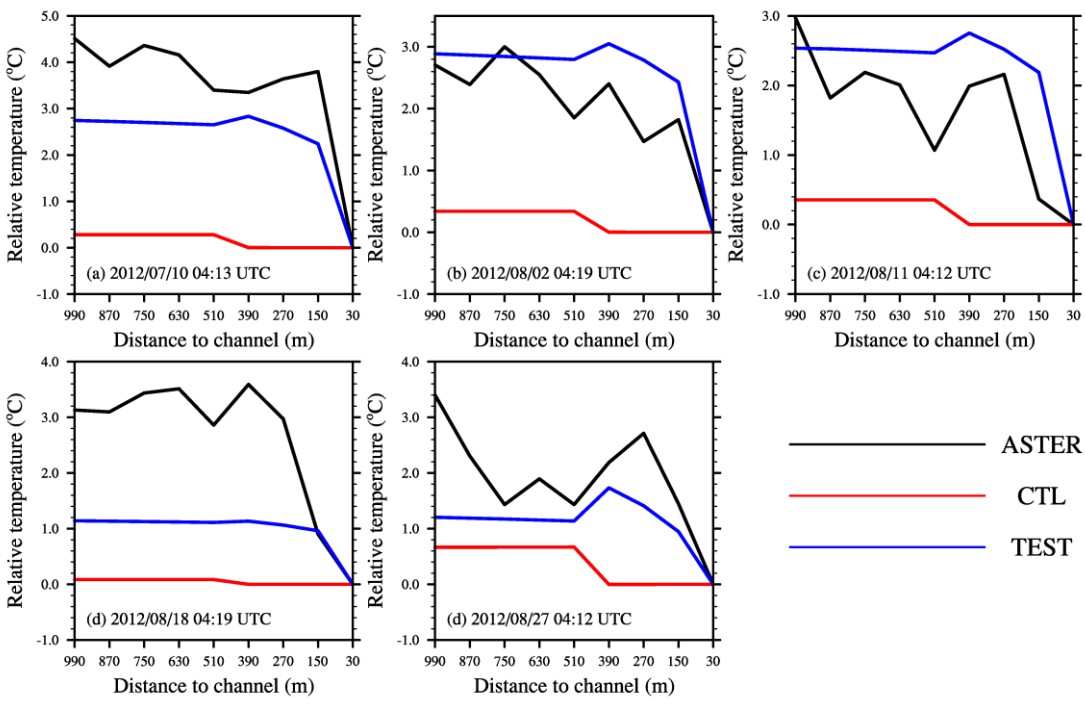

**Figure 8: Relative ground temperature across the left riverbank of the 213 Bridge station from the CTL and TEST simulations and corresponding remote sensing data from five ASTER satellite transit events of (a) 2012/07/10 04:13 UTC, (b) 2012/08/02 04:19 UTC, (c) 2012/08/11 04:12 UTC, (d) 2012/0818 04:19 UTC and (e) 2012/08/27 04:12 UTC.**

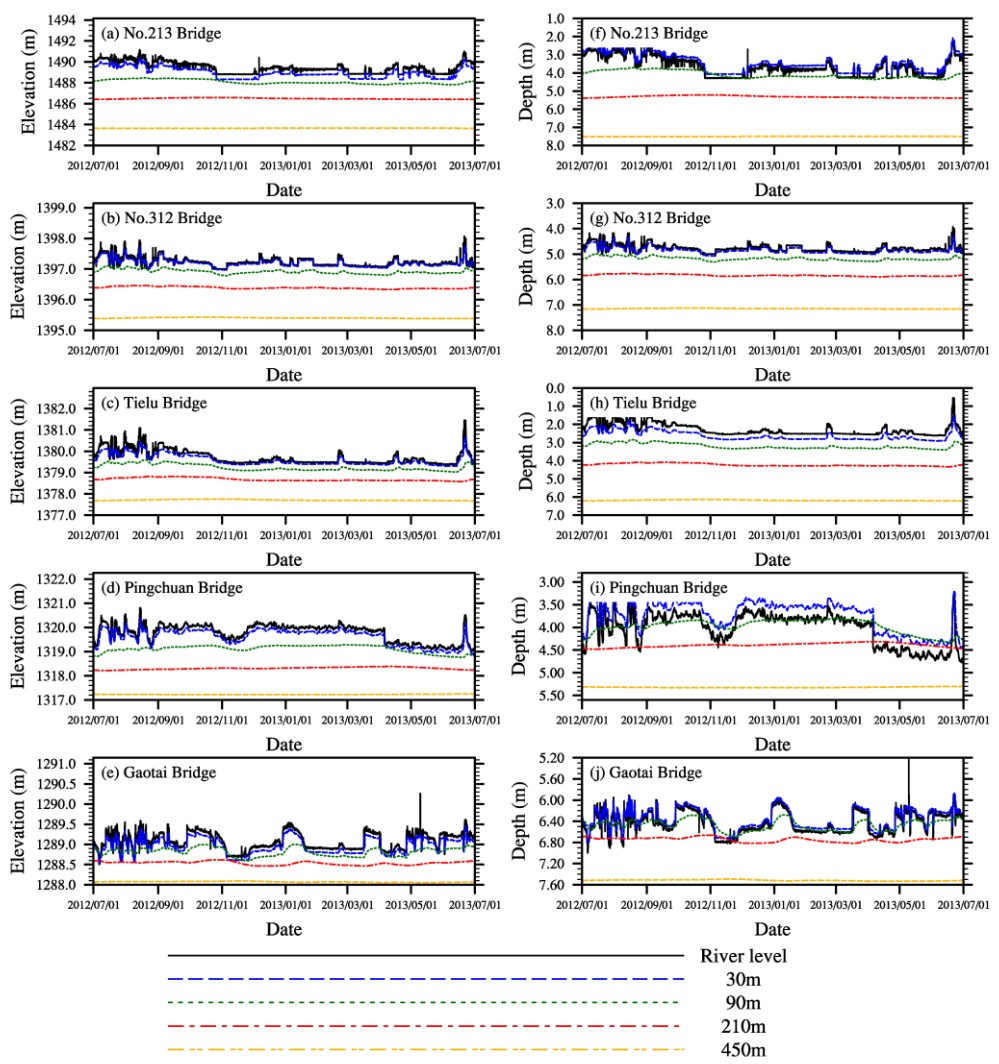

**Figure 9: Time series of simulated (a-e) water head elevations and (f-j) water table depths at 30 m, 90 m, 210 m and 450 m from streams and the observed river water levels at the five left riverbanks of stations at (a, f) 213 Bridge, (b, g) 312 Bridge, (c, h) Tielu Bridge, (d, i) Pingchuan Bridge and (e, j) Gaotai Bridge.**

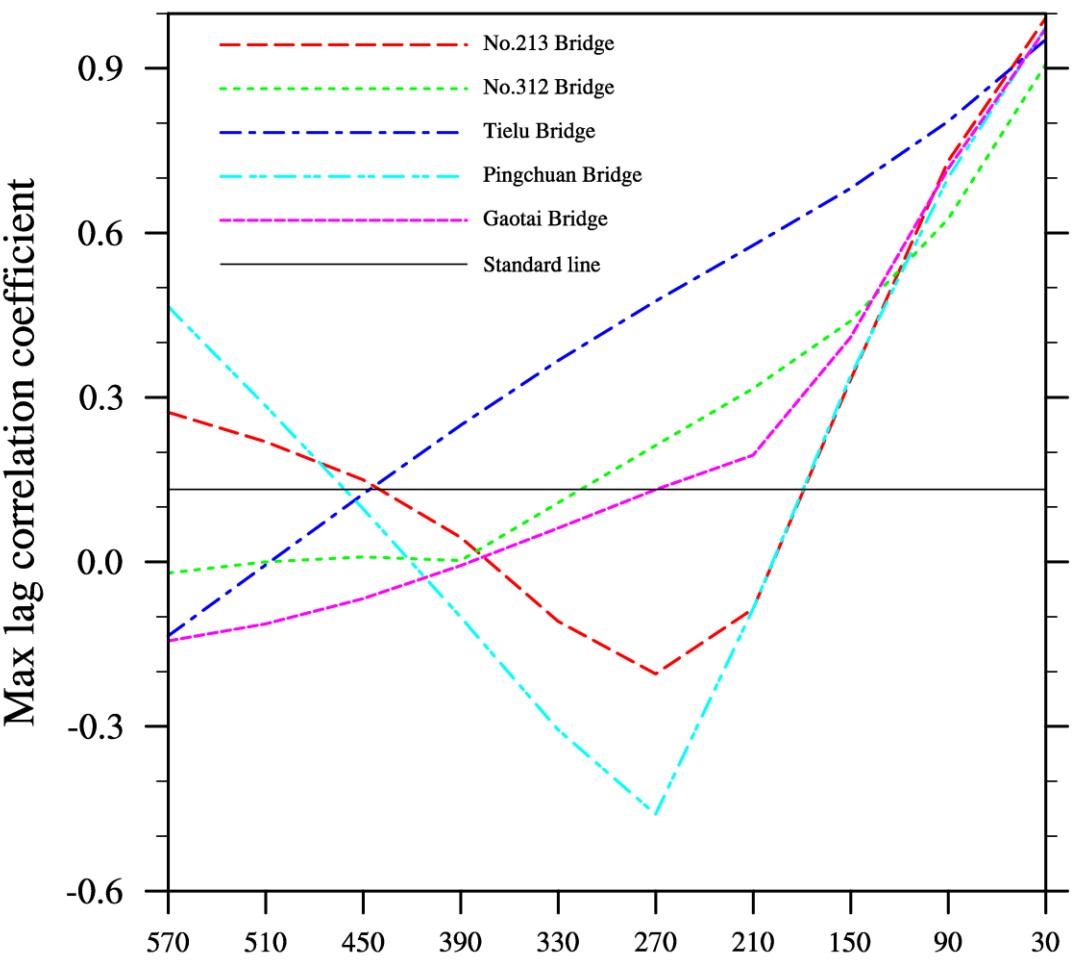

**Figure 10:** Maximum lag correlation coefficients between simulated groundwater tables across the left riverbanks and the river water levels at the five stations, and the standard line representing the value of correlation coefficient passing the Student's t test with a confidence level of 95%.

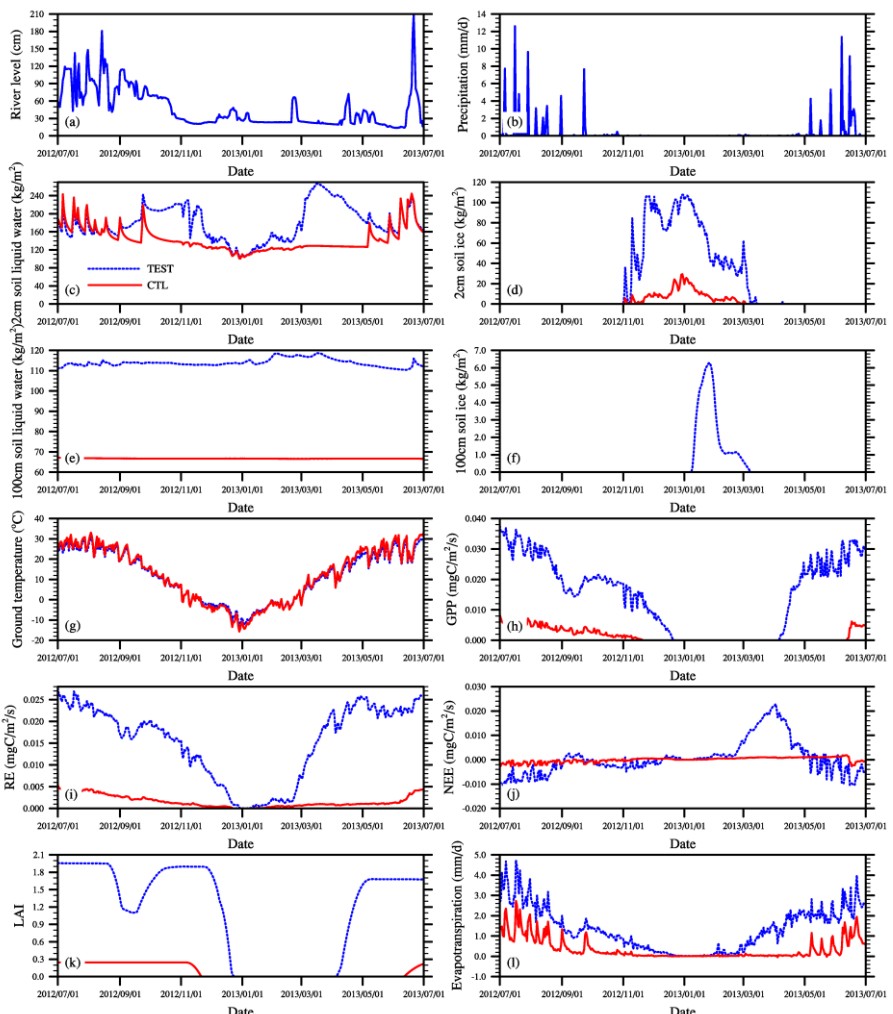

**Figure 11: Time series of area-averaged daily (a) observed river level and (b) observed precipitation, as well as (c) 2-cm soil liquid water, (d) 2-cm soil ice, (e) 100-cm soil liquid water, (f) 100-cm soil ice, (g) ground temperature, (h) gross primary productivity, (i) respiration efficiency, (j) net ecosystem exchange, (k) leaf area index and (l) evapotranspiration predicted by TEST and CTL simulations within 300 m of both sides of the stream at the Tielu Bridge station.**

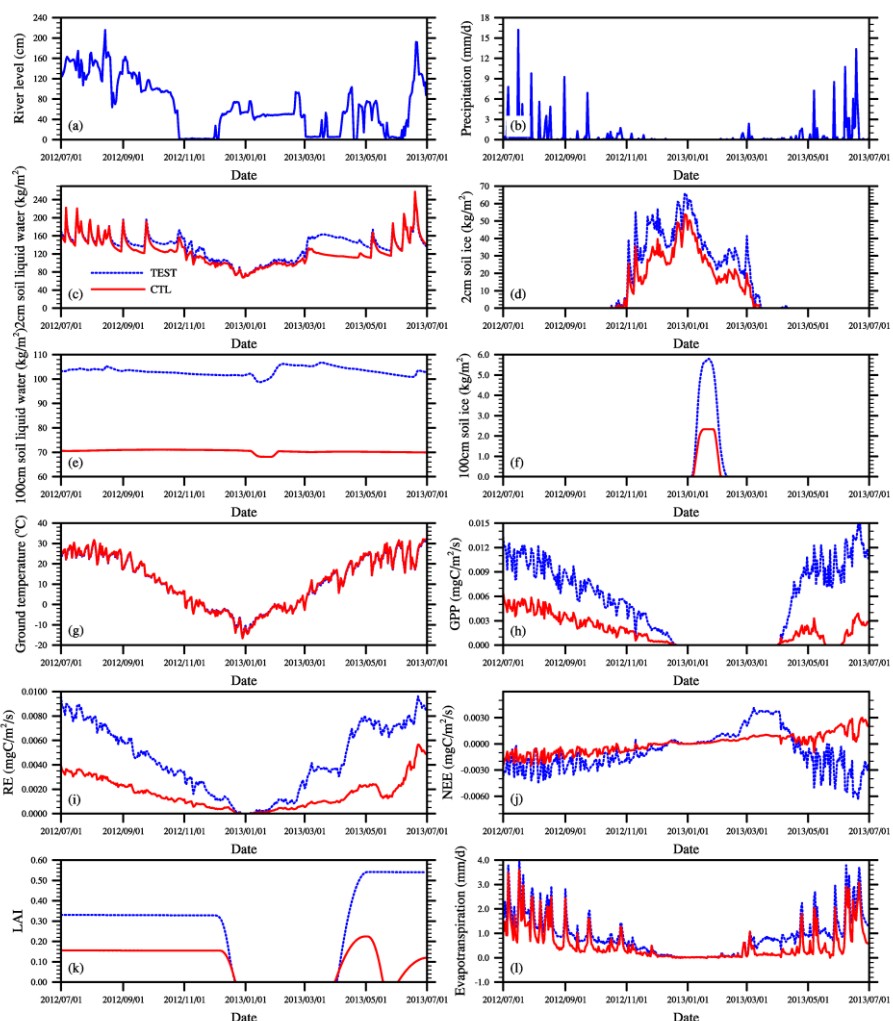

**Figure 12: Time series of area-averaged daily (a) observed river level and (b) observed precipitation, as well as (c) 2-cm soil liquid water, (d) 2-cm soil ice, (e) 100-cm soil liquid water, (f) 100-cm soil ice, (g) ground temperature, (h) gross primary productivity, (i) respiration efficiency, (j) net ecosystem exchange, (k) leaf area index and (l) evapotranspiration predicted by TEST and CTL simulations within 300 m of both sides of the stream at the 213 Bridge station.**

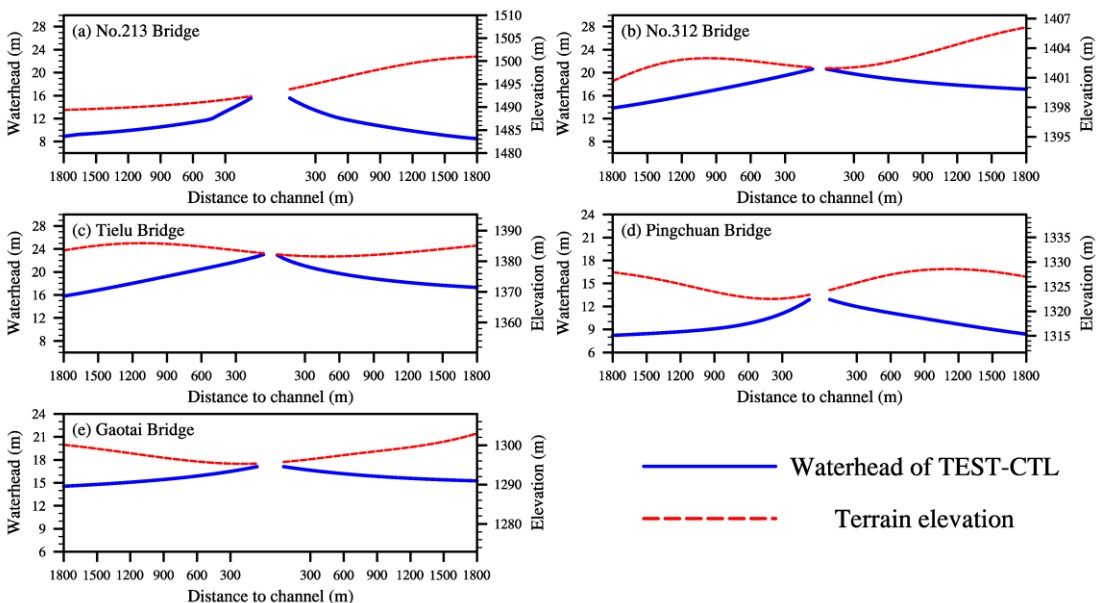

**Figure 13: Differences between annual water heads predicted by TEST and CTL simulations and terrain elevations along the five sections at (a) 213 Bridge, (b) 312 Bridge, (c) Tielu Bridge, (d) Pingchuan Bridge and (e) Gaotai Bridge. The discontinuous parts of the curves represent the river areas.**

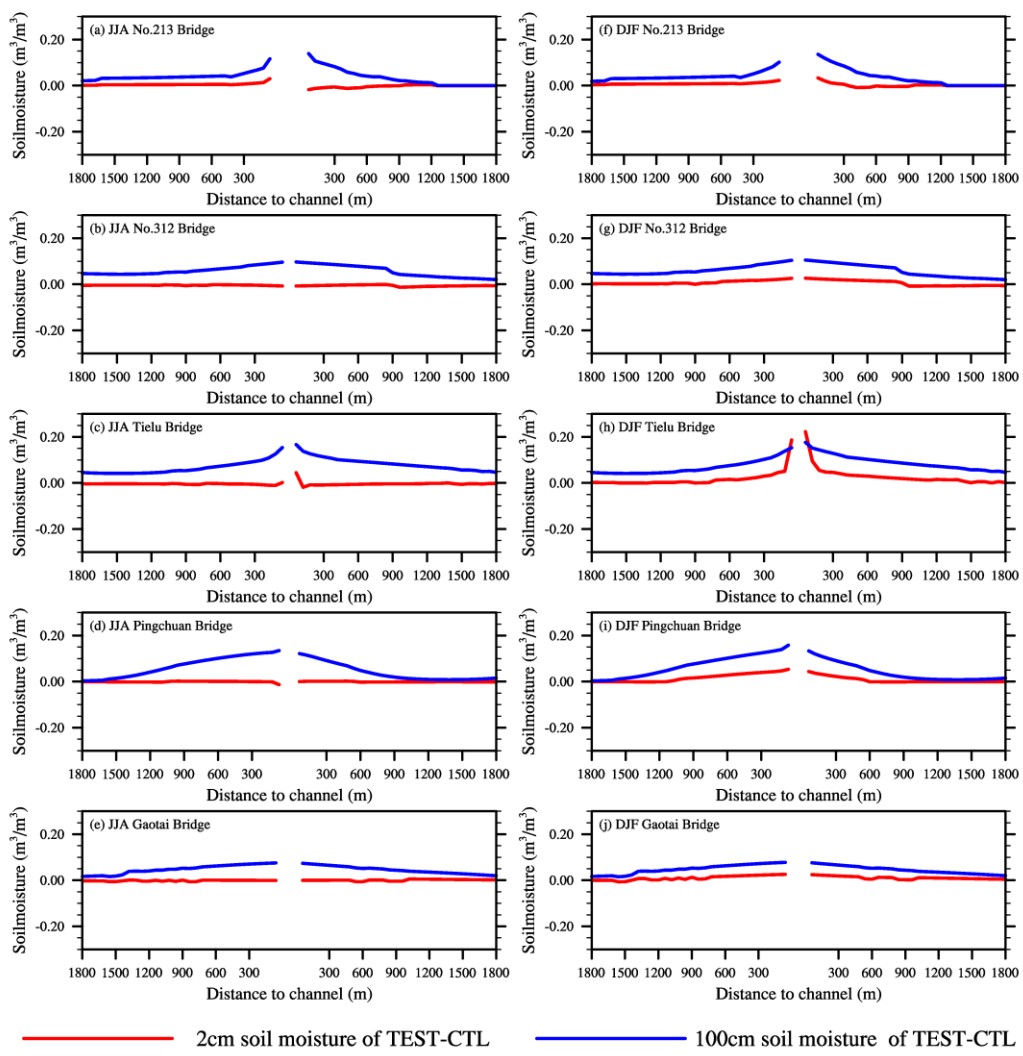

**Figure 14: Differences of (a–e) summer and (f–j) winter soil moisture (both liquid water and ice are included) predicted at depths of 2 cm and 100 cm by TEST and CTL simulations along the five sections at (a and f) 213 Bridge, (b and g) 312 Bridge, (c and h) Tielu Bridge, (d and i) Pingchuan Bridge and (e and j) Gaotai Bridge. The discontinuous parts of the curves represent the river areas.**

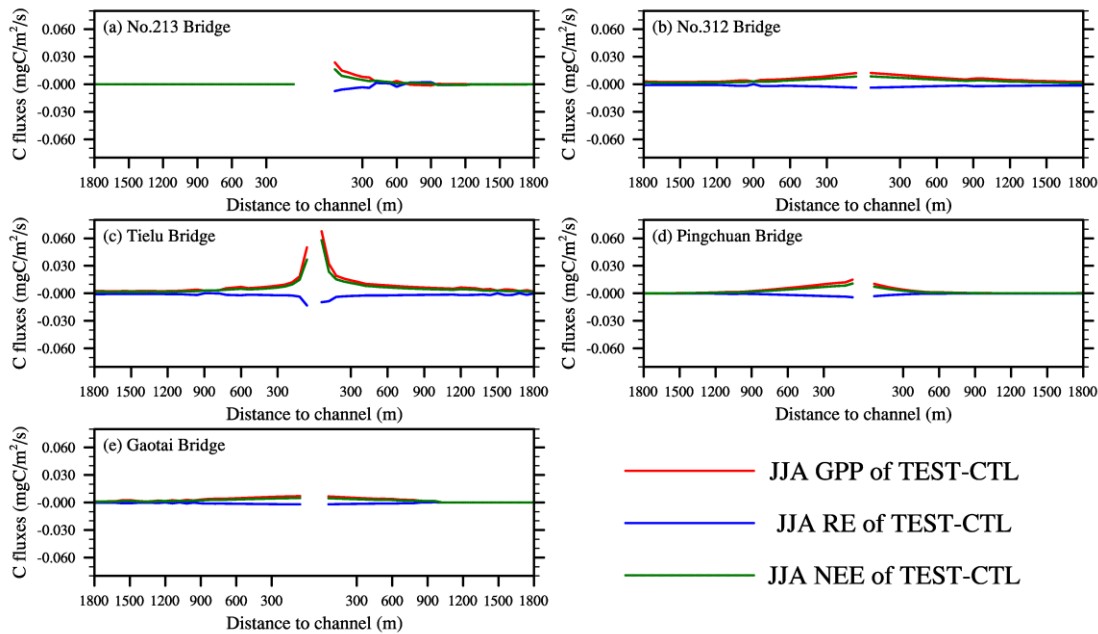

**Figure 15: Differences between gross primary productivity, respiration efficiency and net ecosystem exchange predicted by TEST and CTL simulations during summer along the five sections at (a) 213 Bridge, (b) 312 Bridge, (c) Tielu Bridge, (d) Pingchuan Bridge and (e) Gaotai Bridge. The discontinuous parts of the curves represent the river areas.**

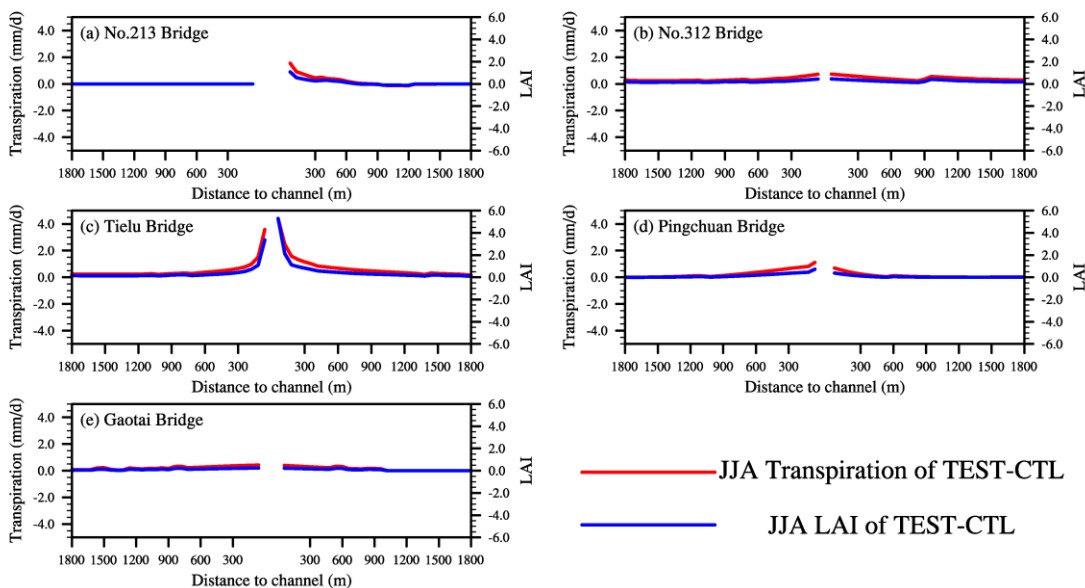

**Figure 16: Differences between canopy transpiration and leaf area index predicted by TEST and CTL simulations during summer along the five sections at (a) 213 Bridge, (b) 312 Bridge, (c) Tielu Bridge, (d) Pingchuan Bridge and (e) Gaotai Bridge. The discontinuous parts of the curves represent the river areas.**

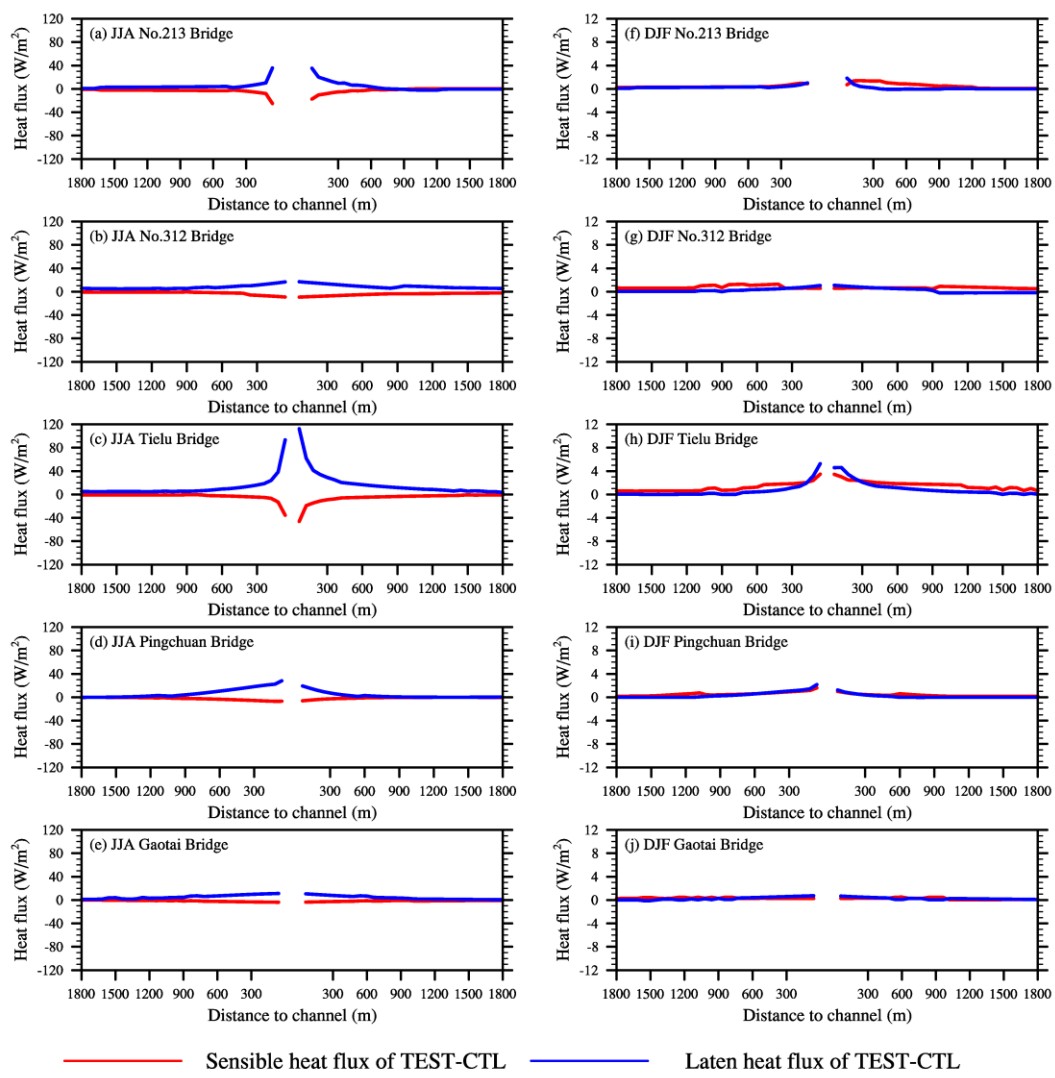

**Figure 17: Differences of (a–e) sensible and (f–j) latent heat fluxes predicted by TEST and CTL simulations along the five sections at (a and f) 213 Bridge, (b and g) 312 Bridge, (c and h) Tielu Bridge, (d and i) Pingchuan Bridge and (e and j) Gaotai Bridge. The discontinuous parts of the curves represent the river areas.**