# Peer review of "Eco-hydrological effects of stream-aquifer water interaction: A case study of the Heihe River Basin, northwestern China"

_Hydrology and Earth System Sciences, 2016_

## Referee Comment (RC1) · Anonymous Referee #1 · 9 Mar 2016

A scheme for stream water - aquifer interaction, originally developed by Di et al (2011), was incorporated into CLM model to simulate the influences of river contribution to aquifers through lateral movement of water. Two simulations with and without activating this scheme were run and compared. Based on the model comparisons the importance of taking the surface water – groundwater interactions into consideration for water, energy and carbon balance models was underscored. Overall, the experiments are interesting and adding such a scheme to CLM could potentially improve models accuracy in areas where groundwater surface water interactions exist especially in riparian areas. However, the paper lacks of clarity in presentation. The objectives were not clearly defined and the approach is somewhat obscure. I believe that the most of

this paper needs to be rewritten and many additional information need to be supplied before it can be considered for publication.

The model was tested in a single case where river is recharging the groundwater. It would be interesting to see how the model response to the other case such as groundwater recharges to the river. As a matter of fact, this could be the case in the study area, where the original CLM in CTL simulations resulted in depth to groundwater levels about 20 m deeper than the observations. However, it is quite difficult to be sure with the provided groundwater observations data. Also, I would be interested in seeing depth to groundwater level values in TEST and CLM simulation to better understand how groundwater levels respond to lateral water movement from the river as well as how close the groundwater to the surface. Groundwater levels were either given as elevations or difference between CTL and TEST simulations but not in depths. I think giving these values as depths would provide more insight in terms of conceptualize the groundwater interactions with the land surface processes.

Even the critically important model parameters were not provided in the paper. I think that the model parameters and initial conditions as well as how these values were determined need to be explained explicitly. Some of the important parameters including, for example, the soil types and parameters in the simulated stations, the vegetation type and their distributions, the specific vegetation parameters and architecture especially root length density distributions were not provided in the paper. Following the paper is somewhat difficult without knowing the model parameters. A table showing the model input parameters would be quite helpful.

Two sets of sensitivity simulations were run. The first one was used to investigate the model responses to river stages and the second one to river bed hydraulic conductivity. In the second sensitivity simulations, it was found that the model results are not sensitive to the hydraulic conductivity of the river bed (Kr). Theoretically, Kr is a parameter that controls the water transfer between the river and the aquifer besides the head difference between them. However, the reason why the model is not sensitive to the Kr

was not discussed in the paper. I think this is important to know and the reason should be discussed in detail and if necessary the additional simulations should be conducted to test if it is a problem related to the numerical scheme or the structure of the model.

An eddy covariance (EC) station was used to validate the model findings. However, some additional information about the station need to be given. For example, the exact location of the station should be placed in Fig. 3. Also, its fetch area needs to be described with the vegetation information. Moreover, EC data were only compared with TEST simulations. I think including the CTL simulations to the same plot would show how the model results improved by adding the stream-aquifer interaction scheme to the model.

In Fig. 7, simulated surface temperature was compared with remotely sensed temperature values. The heat transfer algorithm used in the model can be briefly explained because it would be helpful to understand how the shallower groundwater could alter the surface temperature. Also, I would be curios about to know how the boundary conditions were set up, and how the temperature of the river was treated, used as model forcing or a constant temperature was assigned. Again, I think CTL simulations needs to be included in Fig. 7 as well to show the degree of improvement of the model results. Some specific comments:

1) Abstract could be improved by adding a conclusion sentence. Also, it reads as no validation available in the paper.

2) Introduction part is quite brief and lack of objectives of the paper, which needs to be clearly stated.

3) Model time step definitions should be consistent. Please use either 1800s or 0.5h.

4) The title of section 4.2 is identical with the title of section 4, please add titles properly as necessary.

5) Figures are usually not well presented and explained. For example:

(a) Figure 1c: what the lengths of the boxes represents is not clear. Are these grid cells as described in the caption or they are water heads?

(b) Figure 4: why and how 20 grid cells were used was not explained.

(c) Figure 5, 7: please add CTL simulation results.

(d) Figures 12 to 16: the reason why the left and right sides of the river channel are not symmetrical is not clear. Is it due to the soil type or vegetation? Please clearly provide their distributions.

---

## Referee Comment (RC2) · Anonymous Referee #2 · 12 Apr 2016

Title: Eco-hydrological effects of stream-aquifer water interaction: A case study of the Heihe River Basin, northwestern China Authors: Zeng et al.

Summary: This work presents the impact of river network-aquifer interaction on both sides of the river network using 1D lateral groundwater model. On both sides of the river network groundwater exchange is simulated over a region of 3 km using a pixel resolution of 60 m. For each pixel the vertical column response is simulated using the CLM4.5 model. Results show that the river network has an impact on the saturated and unsaturated zone dynamics in close vicinity of the river network. These variations have an impact on the water, energy and ecological properties of these grid cell.

Overall quality: Reading the title and abstract of this manuscript I was quite enthusiastic

about the content of this work. However, after thoroughly reading the rest of this work I ended up feeling rather disappointed. The authors basically show that incorporating river-groundwater interactions has an impact on the water table and unsaturated zone dynamics. And these variations have impact on the carbon and energy fluxes. As such, the message presented in the abstract does not correspond well with the content of the manuscript. In my comments below I have tried to provide some more detailed information on how to improve this discrepancy. Furthermore, I need to stress the important equations as given by Eqs. 6-8 seem mathematically incorrect (see below). I would like to ask the authors to make sure that these are just typos and that the model was correctly implemented. If this is not the case, the simulations performed in this work need to be redone. That being said, the overall results presented in this work are fine and fit within the scope of HESS. Therefore, in its current form I recommend major changes. These changes are mainly related to the textual content of the manuscript.

General comments:

1) Page 1, lines 15-16 and page 16, line 1 states that stream aquifer interaction processes were incorporated into CLM4.5. I do not agree with this statement. From what I understand from the modelling set up, based on reading the paper, the authors have simulated the hydrological response of 50 pixels of each 60 m wide on both sides of the river and simulated the vertical response of each pixel using CLM 4.5. Furthermore, the response of the river network is not explicitly simulated using CLM4.5 but is externally forced in the model. In the current version for of the manuscript, the authors give the impression as if a major addition was added to the model. I do not believe that this the case while reading the paper. The authors only present 1 dimensional lateral groundwater exchange model, which obtains its water level estimate from CLM4.5.

2) Page 4, lines 11. Generally CLM4.5 is use for large-scale simulation (global/continental) using relatively coarse grid resolution (about 0.1-1 degree). Furthermore, these simulations usually make use of a 2D lateral grid structure, even though the official version of CLM4.5 does not explicitly represent lateral groundwater

flow, but instead the lateral groundwater flux (as estimated using a non-linear reservoir model) is directly moved into the river network. Given this difference in the official version set up and the set up used here (see also previous point) I would suggest to add a section between 2.1 and 2.2 which shows the 1D lateral grid set up up (on both sides of the river network using a high pixel resolution) used here. This will really help improve the readability of the manuscript. E.g. it will then become much easier to understand page 4, lines 9-18.

3) Page 3-4, Eqs. 1-4. The authors present here presents the 1-dimensional lateral groundwater flow equation here with a flexible downstream head boundary condition (i.e. the river network). This model is used to simulated the groundwater response on CLM. In the original version of CLM4.5 a non-linear groundwater reservoir model is used. However, in the manuscript no information is provided, whether this original model was removed in the setup of the authors? Please provide some additional details here (see also comments below).

4) Page 5, line 6 Change "i.e. water . . . 3.8m" to "i.e. water table lies within 3.8m from surface."

5) Page 5, Eq. 6. There is know information provided on what T1 and T2 indicate?

6) Page 5, Eq. 7. Mathematically this is incorrect as the transmissivity is obtained from the groundwater level up to the depth of the bedrock. The summation should therefore not include all 10 layers. Instead if the groundwater level lies within layer i: $(z\_wt - z\_(i,bot))*K\_i$ + summation from layer j=i+1 till layer j=10 of (delta $z\_j*K\_j$). Where $z\_wt$ is the depth of the groundwater table (Eq. 5) and $z\_(i,bot)$ is the bottom level of layer i.

7) Page 5, Eq. 8. After this equation please add: "where, z' = z-3.8.

8) Page 8, line 4-5. The manuscript states that an initial spin-up of 700 years was conducted using the original CLM4.5 model. So without groundwater exchange. This looks very impressive but seems very redundant as well.. Given the resolution of the CLDAS

dataset (0.0625 degrees corresponding to 7.5km), means that all 50 cells on each side of the river receive the same type of input. Without accounting for lateral exchange, basically means that they all give the same results, indicating that the simulations can be performed using a single pixel. I cannot believe that one needs 700 years of spin-up simulations to reach some kind of equilibrium groundwater level. Please provide more information here why this was performed.

9) Page 8, line 15. Please add a line indicating that for the river cell in the middle, no simulations with CLM4.5 were performed. But instead a boundary condition was enforced here.

10) Page 8, lines 12-20. For each of the 50 pixels on both sides of the channel network, did the authors consider elevation variations between the pixels?

11) Page 8, lines 12-20. It is not clear how the control simulations where implemented? Lines 13-15 state the these do not take stream-aquifer interaction into account. It is not clear whether these simulations do account fo lateral groundwater exchange (Eq. 1-4) and how groundwater is removed into the river network (was there some additional boundary condition used?). The results in this manuscript show that there are considerable differences between the CTL and TEST simulations. However, as the current manuscript does not provide much info on how CTL was implemented, it is currently unknown whether how important these difference are (or whether is it just related to the set up of the model).

12) Page 8, line 19. What is the resolution of the MICLCover land cover map used here?

13) Page 8-9, sensitivity experiment 1 and 2.. On page 9, lines 12-13 it is mention that the groundwater table variations are not sensitive to k_r. By directly comparing the chosen values of k_r with those of the saturated lateral hydraulic conductivity value of the surrounding soils Ksoil, one could already made a first impression whether this would have a impact. In case k_r is much larger than Ksoil (as I expect to be the case

here), I do not see a reason why to perform this experiment. As these results were to be expected. Therefore, I would remove these results from the manuscript (this will also reduce the number of figures presented in this work, which is rather large).

14) Page 9, lines 1-2. See comment #8.

15) Page 10, lines 1-3. In my opinion these results just show that the model correctly adjust to changes in the observed surface temperature.

16) Page 10, line 6. Change "good ability" to "reasonable ability".

17) Page 10, lines 9-18 and Fig. 6. The results presented in this figure are heavily depend on the local surface elevation enforced in the model. I would therefore suggest the rescale and plot the difference as with respect to the local surface elevation (y-axes) as function of distance from the channel network (x-axis). This helps to improve the interpretation of this figure.

18) Fig. 7. Note that the legend is a dashed line, while this is not shown in any of the panels.

19) Page 10, lines 17-18. Please mention that the quality of these results are directly influenced by the chosen saturated hydraulic conductivity values, which in this study were chosen a priori and as such not optimized in any kind of manner.

20) Page 12, line 1. Please add ", see Section 4.2.2." after "from a stream".

21) Page 17, lines 14-15. Please remove this statement from the manuscript. This work presents a theoretical study using an extremely high pixel resolution in the direction perpendicular to the river network. Such, resolutions at large scale are infeasible. Even if this would be possible, such an implementation would need many additional model changes not accounted for in the model set up presented in this work.

---

## Author Comment (AC1) · 28 Apr 2016

Please check the attachments for the revised manuscript and our responses to the comments, thanks much for your comments!

Please also note the supplement to this comment:
http://www.hydrol-earth-syst-sci-discuss.net/hess-2016-8/hess-2016-8-AC1-supplement.zip

---

## Author Response (AR1)

**Reviewer #1:**

We thank the reviewer 1 for the helpful comments and suggestions, which are in plain text below. Our response is in **bold** text.

5

10

15

20

A scheme for stream water - aquifer interaction, originally developed by Di et al (2011), was incorporated into CLM model to simulate the influences of river contribution to aquifers through lateral movement of water. Two simulations with and without activating this scheme were run and compared. Based on the model comparisons the importance of taking the surface water – groundwater interactions into consideration for water, energy and carbon balance models was underscored. Overall, the experiments are interesting and adding such a scheme to CLM could potentially improve models accuracy in areas where groundwater surface water interactions exist especially in riparian areas. However, the paper lacks of clarity in presentation. The objectives were not clearly defined and the approach is somewhat obscure. I believe that the most of this paper needs to be rewritten and many additional information need to be supplied before it can be considered for publication.

The model was tested in a single case where river is recharging the groundwater. It would be interesting to see how the model response to the other case such as groundwater recharges to the river. As a matter of fact, this could be the case in the study area, where the original CLM in CTL simulations resulted in depth to groundwater levels about 20 m deeper than the observations. However, it is quite difficult to be sure with the provided groundwater observations data.

Response: Thanks for the comments. As the suggestions, we added some simulations for the case of groundwater recharging river in the sensitivity experiments (page 8, line 28-page 9, line 4; page 9, line 31-page 10, line 18 of the revised manuscript without change tracks), and showed the water table variations in Figure 5.

Also, I would be interested in seeing depth to groundwater level values in TEST and CLM simulation to better understand how groundwater levels respond to lateral water movement from the river as well
as how close the groundwater to the surface. Groundwater levels were either given as elevations or difference between CTL and TEST simulations but not in depths. I think giving these values as depths would provide more insight in terms of conceptualize the groundwater interactions with the land surface processes.

35 Response: Thanks for the comments. As the suggestion, we showed the values of groundwater table depths from CTL, TEST and observation in the Figure 7b, and showed the time series of groundwater table depths from TEST in the Figure 9f-9j. Relevant description and analyses of

1

**the added figures were also settled properly in the manuscript.**

Even the critically important model parameters were not provided in the paper. I think that the model parameters and initial conditions as well as how these values were determined need to be explained explicitly. Some of the important parameters including, for example, the soil types and parameters in the simulated stations, the vegetation type and their distributions, the specific vegetation parameters and architecture especially root length density distributions were not provided in the paper. Following the paper is somewhat difficult without knowing the model parameters. A table showing the model input parameters would be quite helpful.

10

15

Response: Thanks for the comments. As the suggestion, we made a table (table 2 of the manuscript) to show soil type and vegetation type over both sides of the selected five sections, and explained more clearly about how we got the initial conditions (page 9, lines 25-28). To other model parameters, such as the parameters related to atmospheric boundary layer, hydrology, thermodynamics and vegetation (including root length density), the default settings of CLM4.5 were applied. Detailed information about these parameters could be found in the technical description of CLM4.5 (Oleson et al. 2013). The description above was added in the manuscript (page 9, lines 13-20).

Two sets of sensitivity simulations were run. The first one was used to investigate the model responses 20 to river stages and the second one to river bed hydraulic conductivity. In the second sensitivity simulations, it was found that the model results are not sensitive to the hydraulic conductivity of the river bed (Kr). Theoretically, Kr is a parameter that controls the water transfer between the river and the aquifer besides the head difference between them. However, the reason why the model is not sensitive to the Kr was not discussed in the paper. I think this is important to know and the reason 25 should be discussed in detail and if necessary the additional simulations should be conducted to test if it is a problem related to the numerical scheme or the structure of the model.

Response: Thanks for the comments. In fact, the water table is sensitive to the river bed water conductivity Kr in the short-term simulation, while in the long-term simulation the effect of river 30 bed water conductivity is not such significant. To demonstrate this, we plotted the results of short-term (7 days) and long-term (160 days) simulations in Figure 4 and Figure 5. It can be seen that from Figure 4i-4l: As the Kr ranged from 3 m d-1 to 24 m d-1, the time spent by the nearest grid (to river) to get the equilibrium state is shortened from 2 days to 0.5 days. However, after long-term simulation (Figure 4m-4p), the groundwater table depths are similar for all 35 values of Kr. This is because, river bed water conductivity Kr only connects the river and the

nearest model grid (to the river), while the rest of grids (not next to river) are not directly 2

influenced by Kr and are more affected by the lateral hydraulic conductivity K of the riverbank soil (in Eq. (9)). The discussion above were added in the manuscript (page 9, line 31-page 10, line 18) to make the results clearer.

5

10

An eddy covariance (EC) station was used to validate the model findings. However, some additional information about the station need to be given. For example, the exact location of the station should be placed in Fig. 3. Also, its fetch area needs to be described with the vegetation information. Moreover, EC data were only compared with TEST simulations. I think including the CTL simulations to the same plot would show how the model results improved by adding the stream-aquifer interaction scheme to the model.

Response: Thanks for the comments. As the suggestions, the exact location of the station was placed in the Figure 3, and the results from CTL simulation were added to the Figure 5 to show
the improvement of our model. The land cover was also introduced in the manuscript (page 10, lines 26-27).

20

In Fig. 7, simulated surface temperature was compared with remotely sensed temperature values. The heat transfer algorithm used in the model can be briefly explained because it would be helpful to understand how the shallower groundwater could alter the surface temperature. Also, I would be curios about to know how the boundary conditions were set up, and how the temperature of the river was treated, used as model forcing or a constant temperature was assigned. Again, I think CTL simulations needs to be included in Fig. 7 as well to show the degree of improvement of the model results.

25

30

Response: Thanks for the comments. As the suggestions, we gave a brief introduction of the heat transfer algorithm (including the boundary condition) of CLM (page 7, lines 9-26) and explained why the temperature was affected by the stream-aquifer water interaction. Currently, the river temperature and the horizontal heat transfer are not included, but will be incorporated to our model in the future. The CTL simulations were added in Figure 8 to show the degree of

improvement of the model results.

Some specific comments:

available in the paper.

1) Abstract could be improved by adding a conclusion sentence. Also, it reads as no validation

35

Response: Thanks for the comments. We have added the sentences about the model validation

**(page 1, lines 15-16) and conclusions (page 1, lines 26-27) as suggestions.**

2) Introduction part is quite brief and lack of objectives of the paper, which needs to be clearly stated.

**5 Response: Thanks for the comments. We have enriched the introduction (page 1, line 29-page 2, line 3) and added the objectives of the manuscript (page 2, lines 29-33) as the suggestions.**

3) Model time step definitions should be consistent. Please use either 1800s or 0.5h.

**10 Response: Thanks for the comments. We unified the time step to 1800 s in the manuscript.**

4) The title of section 4.2 is identical with the title of section 4, please add titles properly as necessary.

**15 Response: Thanks for the comments. We changed the title of section 4 to "Results".**

5) Figures are usually not well presented and explained. For example:

(a) Figure 1c: what the lengths of the boxes represents is not clear. Are these grid cells as described in the caption or they are water heads?

20

Response: Thanks for the comments. We have modified the figure based on the suggestion. The dash lines represent the water heads in the new revised figure.

(b) Figure 4: why and how 20 grid cells were used was not explained.

**25**

**Response:** Thanks for the comments. We plotted water tables of all the grid cells in the revised Figure 4 and Figure 5 instead of showing only 20 grids.

(c) Figure 5, 7: please add CTL simulation results.

30

**Response: Thanks for the comments. We have added the CTL simulation results to these figures as the suggestion.**

(d) Figures 12 to 16: the reason why the left and right sides of the river channel are not symmetrical isnot clear. Is it due to the soil type or vegetation? Please clearly provide their distributions.

**Response: Thanks for the comments. The asymmetrical effects was mainly produced by the**

asymmetrical topography. We plotted the terrain elevations of both sides of the 5 sections in Figure 13 and gave an explanation for the asymmetric effects (page 14, lines 9-16).

5

**Reviewer #2:**

We thank the reviewer 2 for the helpful comments and suggestions, which are in plain text below. Our response is in **bold** text.

5

Title: Eco-hydrological effects of stream-aquifer water interaction: A case study of the Heihe River Basin, northwestern China Authors: Zeng et al.

10

Summary: This work presents the impact of river network-aquifer interaction on both sides of the river network using 1D lateral groundwater model. On both sides of the river network groundwater exchange is simulated over a region of 3 km using a pixel resolution of 60 m. For each pixel the vertical column response is simulated using the CLM4.5 model. Results show that the river network has an impact on the saturated and unsaturated zone dynamics in close vicinity of the river network. These variations have an impact on the water, energy and ecological properties of these grid cell.

15

35

Overall quality: Reading the title and abstract of this manuscript I was quite enthusiastic about the content of this work. However, after thoroughly reading the rest of this work I ended up feeling rather disappointed. The authors basically show that incorporating river-groundwater interactions has an impact on the water table and unsaturated zone dynamics. And these variations have impact on the

- 20 carbon and energy fluxes. As such, the message presented in the abstract does not correspond well with the content of the manuscript. In my comments below I have tried to provide some more detailed information on how to improve this discrepancy. Furthermore, I need to stress the important equations as given by Eqs. 6-8 seem mathematically incorrect (see below). I would like to ask the authors to make sure that these are just typos and that the model was correctly implemented. If this is not the
- 25 case, the simulations performed in this work need to be redone. That being said, the overall results presented in this work are fine and fit within the scope of HESS. Therefore, in its current form I recommend major changes. These changes are mainly related to the textual content of the manuscript.
- 30 General comments:

1) Page 1, lines 15-16 and page 16, line 1 states that stream aquifer interaction processes were incorporated into CLM4.5. I do not agree with this statement. From what I understand from the modelling set up, based on reading the paper, the authors have simulated the hydrological response of 50 pixels of each 60 m wide on both sides of the river and simulated the vertical response of each pixel using CLM 4.5. Furthermore, the response of the river network is not explicitly simulated using CLM4.5 but is externally forced in the model. In the current version for of the manuscript, the authors

give the impression as if a major addition was added to the model. I do not believe that this the case

while reading the paper. The authors only present 1 dimensional lateral groundwater exchange model, which obtains its water level estimate from CLM4.5.

5

Response: Thanks for the comments. As the suggestion, we changed the way of expression from "incorporating stream-aquifer interaction scheme to CLM and model development" to "combining the two models to investigate the effects of stream-aquifer water interaction" in the abstract, introduction and throughout the manuscript.

10

In fact, the two models are two-way coupled. That means, besides the presenting of one-dimensional lateral groundwater exchange model which obtains its water level (and some other parameters) estimate from CLM4.5, we also modified the simulated groundwater table and aquifer water storage of CLM4.5 based on the output of the lateral groundwater exchange model. We took advantages of both the models. It is not a major modification to CLM but can be seen as a convenient and effective way to achieve our scientific goals.

- 15 2) Page 4, lines 11. Generally CLM4.5 is use for large-scale simulation (global/continental) using relatively coarse grid resolution (about 0.1-1 degree). Furthermore, these simulations usually make use of a 2D lateral grid structure, even though the official version of CLM4.5 does not explicitly represent lateral groundwater flow, but instead the lateral groundwater flux (as estimated using a non-linear reservoir model) is directly moved into the river network. Given this difference in the official version
- 20 set up and the set up used here (see also previous point) I would suggest to add a section between 2.1 and 2.2 which shows the 1D lateral grid set up up (on both sides of the river network using a high pixel resolution) used here. This will really help improve the readability of the manuscript. E.g. it will then become much easier to understand page 4, lines 9-18.
- Response: Thanks for the comments. As the suggestion, we added a section of "Configuration of CLM4.5 for simulation over riverbank" (page 3, line 18-page 4, line 6 of the revised manuscript without change tracks) to introduce how we set up the model and prepared the surface dataset which is the most important in riverbank simulation. Furthermore, the subsurface runoff scheme in CLM4.5 was turned off because it was not suitable in the fine-scale modeling and replaced by the groundwater lateral flow in stream-aquifer interaction scheme, which was the explicit representation of the subsurface process (page 4, lines 2-6).

3) Page 3-4, Eqs. 1-4. The authors present here presents the 1-dimensional lateral groundwater flow equation here with a flexible downstream head boundary condition (i.e. the river network). This model is used to simulated the groundwater response on CLM. In the original version of CLM4.5 a non-linear groundwater reservoir model is used. However, in the manuscript no information is provided, whether this original model was removed in the setup of the authors? Please provide some

additional details here (see also comments below).

Response: Thanks for the comments. The flexible downstream head boundary condition was only used when running the stream-aquifer water interaction module and did not directly connect to CLM4.5. All the vertical biogeophysical and biogeochemical processes of CLM4.5 5 was retained because they were not scale-dependent and could be used in any resolution if the corresponding surface dataset was set properly. To the non-linear groundwater reservoir model of original CLM4.5, the vertical water exchange scheme between soil and aquifer was not modified. However, as referred above, the subsurface runoff of the original CLM4.5 was turned 10 off in the model because it was not fit for the fine-scale modeling and was replaced by our lateral groundwater exchange model. The Relevant discussions were added in the new section 2.2 of "Configuration of CLM4.5 for simulation over riverbank" in the manuscript (page 3, line 18-page 4, line 6).

15 4) Page 5, line 6 Change "i.e. water : : : 3.8m" to "i.e. water table lies within 3.8m from surface."

**Response: Thanks for the comments. We modified the sentence as the suggestion.**

5) Page 5, Eq. 6. There is know information provided on what T1 and T2 indicate?

20

**Response: Thanks for the comments. We added related information about $T_1$ and $T_2$ in the appropriate position (page 5, lines 17-18).**

- 6) Page 5, Eq. 7. Mathematically this is incorrect as the transmissivity is obtained from the groundwater level up to the depth of the bedrock. The summation should therefore not include all 10 25 layers. Instead if the groundwater level lies within layer i:  $(z_wt - z_i,bot)$ \*K\_i + summation from layer j=i+1 till layer j=10 of (delta  $z_j * K_j$ ). Where  $z_w$  is the depth of the groundwater table (Eq. 5) and z\_(i,bot) is the bottom level of layer i.
- Response: Thanks for the comments. We are sorry for this mistake and corrected it in the 30 manuscript (page 5, line 15). We ensure that it is only a slip of typing. The model code is correct.

7) Page 5, Eq. 8. After this equation please add: "where, z' = z-3.8.

**Response: Thanks for the comments. We added the sentence as the suggestion. 35**

8) Page 8, line 4-5. The manuscript states that an initial spin-up of 700 years was conducted using the 8

original CLM4.5 model. So without groundwater exchange. This looks very impressive but seems very redundant as well. Given the resolution of the CLDAS dataset (0.0625 degrees corresponding to 7.5km), means that all 50 cells on each side of the river receive the same type of input. Without accounting for lateral exchange, basically means that they all give the same results, indicating that the simulations can be performed using a single pixel. I cannot believe that one needs 700 years of spin-up simulations to reach some kind of equilibrium groundwater level. Please provide more information here why this was performed.

5

30

Response: Thanks for the comments. Although the resolution of atmospheric forcing dataset is
coarse, the topographic, land cover, soil datasets for making CLM surface dataset are fine
(ASTER Dem Dataset with 30-m, MICLCover with 1-km, HiWATER Land Cover Map with
30-m and China Soil Characteristics Dataset with 1-km). So we think it is necessary for the
spin-up over all grids. The choice of 700 "spin-up" years was based on the user's guide of CLM
(Chapter 4 of Kluzek 2013) showing that when the biogeochemistry carbon-nitrogen module of
CLM is turned on (it is the case of this study), the model should be at least run for 700 years to
get a steady state because the magnitudes of carbon and nitrogen fluxes are very small (Oleson et al. 2013). The discussion above was added in the manuscript (page 8, lines 25-28; page 9, lines 13-20).

20 9) Page 8, line 15. Please add a line indicating that for the river cell in the middle, no simulations with CLM4.5 were performed. But instead a boundary condition was enforced here.

**Response: Thanks for the comments. We added the sentence as the suggestion.**

25 10) Page 8, lines 12-20. For each of the 50 pixels on both sides of the channel network, did the authors consider elevation variations between the pixels?

Response: Thanks for the comments. Certainly we took the elevation variations into consideration, for it is the major control of the groundwater lateral flow. We got the high-resolution elevation data from ASTER Dem Dataset with 30-m resolution.

11) Page 8, lines 12-20. It is not clear how the control simulations where implemented? Lines 13-15 state the these do not take stream-aquifer interaction into account. It is not clear whether these simulations do account for lateral groundwater exchange (Eq. 1-4) and how groundwater is removed
into the river network (was there some additional boundary condition used?). The results in this manuscript show that there are considerable differences between the CTL and TEST simulations. However, as the current manuscript does not provide much info on how CTL was implemented, it is

currently unknown whether how important these difference are (or whether is it just related to the set up of the model).

Response: Thanks for the comments. The control simulations took the groundwater lateral flow
into account because in this study we focused on the effects of stream-aquifer water interaction, but not the groundwater lateral flow. The only difference of CTL from TEST is that the water exchange between stream and aquifer was set to zero (flexible boundary condition). We added this information into the manuscript (page 9, lines 7-9).

10 12) Page 8, line 19. What is the resolution of the MICLCover land cover map used here?

Response: Thanks for the comments. The resolution of MICLCover land cover is 1-km. The divide of land cover types of MICLCover is similar to CLM. However we also referred the HiWATER Land Cover Map (30-m resolution) when making the surface dataset. We added this information in the manuscript (page 9, lines 13-20).

13) Page 8-9, sensitivity experiment 1 and 2. On page 9, lines 12-13 it is mention that the groundwater table variations are not sensitive to k\_r. By directly comparing the chosen values of k\_r with those of the saturated lateral hydraulic conductivity value of the surrounding soils Ksoil, one could already
made a first impression whether this would have a impact. In case k\_r is much larger than Ksoil (as I expect to be the case here), I do not see a reason why to perform this experiment. As these results were to be expected. Therefore, I would remove these results from the manuscript (this will also reduce the number of figures presented in this work, which is rather large).

Response: Thanks for the comments. It is right that k\_r is several times larger than Ksoil. However, the k\_r is still matter when the simulation time is short. To show this (as well as to meet the comments from another reviewer), we added the results from short-term (7 days) sensitivity experiments in the Figure 4a-4d and 4i-4l. They revealed that the river bed water conductivity is more important in the controlling of short-term water table variation than the controlling of long-term water table equilibrium. The discussion was added in the manuscript

14) Page 9, lines 1-2. See comment #8.

(page 9, line 31-page 10, line 18).

15

35 Response: Thanks for the comments. The reason of 700 years spin-up was explained in the response to comment #8.

15) Page 10, lines 1-3. In my opinion these results just show that the model correctly adjust to changes in the observed surface temperature.

Response: Thanks for the comments. As the suggestion, we changed the sentence of the explanation for Figure 6a.

16) Page 10, line 6. Change "good ability" to "reasonable ability".

**Response: Thanks for the comments. We changed the words as the suggestion.**

10

5

17) Page 10, lines 9-18 and Fig. 6. The results presented in this figure are heavily depend on the local surface elevation enforced in the model. I would therefore suggest the rescale and plot the difference as with respect to the local surface elevation (yaxes) as function of distance from the channel network (x-axis). This helps to improve the interpretation of this figure.

15

**Response: Thanks for the comments. We plotted the figure (Figure 7c) as the review's suggestion over the Gaotai Bridge where most water wells were displayed.**

18) Fig. 7. Note that the legend is a dashed line, while this is not shown in any of the panels.

20

25

**Response: Thanks for the comments. We modified the Figure 8 to make it clearer.**

19) Page 10, lines 17-18. Please mention that the quality of these results are directly influenced by the chosen saturated hydraulic conductivity values, which in this study were chosen a priori and as such not optimized in any kind of manner.

Response: Thanks for the comments. As the suggestion, we added this statement in the manuscript (page 11, lines 20-22).

20) Page 12, line 1. Please add ", see Section 4.2.2." after "from a stream".

**Response: Thanks for the comments. As the suggestion, we changed the words as the suggestion.**

21) Page 17, lines 14-15. Please remove this statement from the manuscript. This work presents a
theoretical study using an extremely high pixel resolution in the direction perpendicular to the river network. Such, resolutions at large scale are infeasible. Even if this would be possible, such an implementation would need many additional model changes not accounted for in the model set up

presented in this work.

5

Response: Thanks for the comments. We deleted these statements as the suggestion. Maybe in the future we can summarize the findings over the high resolution pixels and come up with some parameterizations to make CLM being able to simulate the effects of stream-aquifer water interaction over large-scale. Thank you!

12

**Eco-hydrological effects of stream-aquifer water interaction: A case study of the Heihe River Basin, northwestern China**

Yujin Zeng1, 2, Zhenghui Xie1, Yan Yu3, Shuang Liu1, 2, Linying Wang1, 2, Binghao Jia1, Peihua Qin1, Yaning Chen4

5

1State Key Laboratory of Numerical Modeling for Atmospheric Sciences and Geophysical Fluid Dynamics, Institute of Atmospheric Physics, Chinese Academy of Sciences, Beijing, 100029, China 2College of Earth Science, University of Chinese Academy of Sciences, Beijing 100049, China

3Zhejiang Institute of Meteorological Sciences, Hangzhou, 310008, China

10 4Key Laboratory of Oasis Ecology and Desert Environment, Xinjiang Institute of Ecology and Geography, Chinese Academy of Sciences, Urumqi, 830011, China

Correspondence to: Zhenghui Xie (zxie@lasg.iap.ac.cn)

- 15 Abstract. A scheme describing the process of stream-aquifer interaction wasand the land model CLM4.5 werewas combined with the land model CLM4.5 applied\_\_incorporated into the land model CLM4.5 to-\_investigateinvestigate the effects of stream water conveyance over riparian banks on ecological and hydrological processes. Two groups of simulations for five typical river cross-sections in the middle reaches of the arid zone Heihe River Basin were conducted. The comparisons between the simulated results and the measurements from water wells, fluxnet station and remote sensing data showed good
- 20 performance good skills of the coupled model. The simulated riparian groundwater table at a propagation distance of less than 1 km followed the intra-annual fluctuation of the river water level, and the correlation was excellent ( $R^2 = 0.9$ ) between the river water level and the groundwater table at the distance 60 m from the river. The correlation rapidly decreased as distance increased. In response to the variability of the water table, soil moisture at deep layers also followed the variation of river water level all year, while soil moisture at the surface layer was more sensitive to the river water level in the drought
- 25 season than in the wet season. With increased soil moisture, the average gross primary productivity and respiration of riparian vegetation within 300 m from the river at a typical section of the river increased by approximately 0.03 mg C m-2 s-1 and 0.02 mg C m-2 s-1, respectively, in the growing season. Consequently, the net ecosystem exchange increased by approximately 0.01 mg C m-2 s-1, and the evapotranspiration increased by approximately 3 mm d-1. Furthermore, the length

of the growing season of riparian vegetation also increased by 2–3 months due to the sustaining water recharge from the river.– Overall, the stream-aquifer water interaction plays an essential role in the controlling of riparian hydrological and ecological processes.

**5 1 Introduction**

Water is indispensable for human society and eco-hydrological system (Milly et al. 2005; Ouyang et al. 2003; Shen and; Chen 2010; Zhao and; Cheng 2002). Among variety kinds of water resources, aquifer water and stream water, which constitute more than 30% of the freshwater storage, are key factors in hydrological cycle (Chen and; Xie 2010; Sch är et al. 1999; Xie et al. 2014; Yu et al. 2014). The aquifer water usually , with its table spatiotemporal fluctuations, acts as a water 10 buffer reservoir to the ecological and hydrological system (Fan 2015; Tsur and; Graham-Tomasi 1991). From the perspective of time, unconfined aquifer stores water resource by plenty recharge from soil infiltration and precipitation lin humid season, aquifer water can store the excess rainfall, and in arid season, it reversely recharges discharges water theto wet root-zone soil and sustains the ecosystem above by upwards capillary flux (Nepstad et al. 1994). The stream is also very important in the eco-hydrological system. It From the perspective of space, the aquifer water lateral flow-continuously 15 transports water from humid-or ridge regions to arid-or valley regions and supports the ecosystem in the lower-reach area (Contreras et al. 2011; Jobbagy et al. 2011). If considering the topography and river network distribution from catchment scale, water interaction between aquifer and river stream is an essential factor to explain the pattern of groundwater table level, soil moisture, evapotranspiration, water balances, vegetation types and status, and even micro scale climate along with the river bank (Constantz 1998; Lehr et al. 2015; McNamara et al. 2005). In wet region or season, rainfall or snow 20 melting can uplift groundwater table head, make it higher than vicinal river level and sustain base flow (Arnold et al. 2000). While in arid region or season when river level is higher than local groundwater table head, channel water, which is conveyed from upstream areas, will be transmitted from river to unconfined aquifer in the downstream areas by lateral discharge (Scanlon et al. 2002). This process is the key factor to maintain the riparian ecosystem over arid and semi-arid areas and that is why we can see lush forests line the river corridors but shrubs and grasses cover the plateaus (Sheldon et 25 al. 2010). If we decide to implement some measures (e.g. artificial water conveyance) to maintain the subsistence of the

riparian ecosystems adapting to climate change and heavier human water withdraw over arid or semi-arid regions where vegetation growths in these regions depend on the water transported from river lateral recharge rather than precipitation, to understand and quantify this effects of the water interaction between stream and unconfined aquifer could not be more important (Baskaran et al. 2009).

[revised manuscript text omitted]

conducted done for CLM4.5 to make the model suitable for tin the one-dimensional 1D and fine-scale simulation.

2.2 Configuration of CLM4.5 for simulation over riverbank

5

10

As an example, t∓o a certain riverbank on one side of a cross-sectionthe stream river, achieve these, we first made the one-dimensional-(1×50 in this study)) –surface dataset —used in CLM4.5 simulation for the one-side riverbank using
 surfacedata-generated tool (Kluzek 2012Reference))... The –schematic diagram for these one-dimensional1D grids of the surface dataset was shown in Figure 1c (m=50)... AAat this time, the longitude and latitude values of each grid wereas set ghosen arbitrarily because they would be were-modified later)the... and Tthen we changed the longitude and latitude of each grid to make themit represent eorresponding to the real location of each site over the riverbank. Next, we replaced each grid's the elevation, terrain slope, maximum fractional saturated area, land cover types (bare ground,- vegetation)

- 20 lakes, etc., etc.) and soil types (percentage of clay, silt, -and-sand and soil organic matter)-of the surface dataset of over each grid of the surface dataset with high-resolution-dataset of ASTER Dem Dataset, (Hirano Akira-et al. 2003; Li et al. 2011), MICLCover, (Ran et al. 2012), HiWATER Land Ceover Mmap and HJ-CCD limage Ddataset (LiZhongLiao et al. 2013), and China Soil Characteristics Dataset (Shangguan et al. 2012)-according to the real. At last, the subsurface runoff scheme in CLM4.5 -(as estimated using a non-linear reservoir model)-was turned off because it was not suitable in the fine-scale
- 25 modeling and would be replaced by the groundwater lateral flow in stream-aquifer interaction scheme (described in the Sect. 17

|----|-------|-------|------|

|-------------------|-------|-------|------|

2.3, which was the explicit representation of the subsurface process) subsurface runoff process. All the vertical biogeophysical and biogeochemical processes of CLM4.5 was retained because they wereare not scale-dependent and could an be used in any resolution if the correspondinging surface dataset was set properly. We Then we modified the code of CLM4.5 so that all the calculation using the latitude and longitude of grid (most of them are related to radiation) applied the location information of the stream. each ... And then ...

**2.32 Scheme for stream-aquifer interaction and its implementation into CLM4.5Scheme for stream-aquifer interaction and its implementation into CLM4.5**

The stream-aquifer water interaction scheme (including groundwater lateral flow) developed by Di et al. (2011) was combined with incorporated into CLM4.5 (the combined model wasand called CLM\_RIV). We first describe the new model briefly as follows. Based on Darcy's law and the Dupuit approximation (Bear, 1972), the lateral flow between a river and the neighboring groundwater can be expressed as:

$$R(x,t) = \frac{\partial Q}{\partial x} = \frac{\partial}{\partial x} \left( T(x,t) \frac{\partial h(x,t)}{\partial x} \right), x > 0, t \ge 0,$$
(1)

while the corresponding initial and boundary conditions are expressed as:

$$h(x,0) = h_0(x), \tag{2}$$

20

5

10

$$h(0,t) = h_{river}(t), \qquad (3)$$

where *x* (L) is the perpendicular distance from the point on a bank to the river channel, *t* (T) is time, R(x,t) (L/T) is the lateral groundwater recharge (or discharge) rate at point *x* and time *t*, Q [L2/T] is the lateral flow discharge, T(x,t) (L2/T) is the lateral flow transmissivity, h(x,t) (L) is the groundwater table elevation,  $h_0(x)$  (L) is the initial groundwater table elevation and  $h_{river}(t)$  (L) is the river water level, as shown in Figures 1a and 1b. If the river water level is higher in elevation than its neighboring groundwater table (as shown in Figure 1a), R(x,t) is greater than zero and the local aquifer discharges to the stream.

To combine incorporate the stream-aquifer interaction scheme with into-CLM4.5, the continuity Eq. (1) should be

**带格式的:**字体:(中文)+中文正文(宋体),字体颜 色:红色 discretized over a modelthe one-dimensional1D grids of the surface dataset of CLM4.5 (Figure 1c) grid and each variable should be linked to CLM4.5. Applying the zero-flux boundary condition to the outermost grid of the simulation domain, the discrete formation of Eq. (1) can be written as:

$$\begin{cases} R_{1,n} = \frac{\frac{T_{0,n} + T_{1,n}}{2} \times \frac{h_m - h_{1,n}}{\Delta x/2}}{\Delta x} - \frac{\frac{T_{1,n} + T_{2,n}}{2} \times \frac{h_{1,n} - h_{2,n}}{\Delta x}}{\Delta x} \\ R_{i,n} = \frac{\frac{T_{i-1,n} + T_{i,n}}{2} \times \frac{h_{i-1,n} - h_{i,n}}{\Delta x}}{\Delta x} - \frac{\frac{T_{i,n} + T_{i+1,n}}{2} \times \frac{h_{i,n} - h_{i+1,n}}{\Delta x}}{\Delta x}, 2 \le i \le m - 1, (4) \\ R_{m,n} = \frac{\frac{T_{m-1,n} + T_{m,n}}{2} \times \frac{h_{m-1,n} - h_{m,n}}{\Delta x}}{\Delta x}}{\Delta x} \end{cases}$$

5 where *i* is the number of the grid that is successively added with the increasing distance from grid to channel (Figure 1c), *m* is the farthest grid from the river channel in the model (i.e., the outermost grid of the simulation domain), *n* is the number of the time step,  $R_{i,n}$  (L/T) is the lateral groundwater recharge (or discharge) rate of grid *i* at the nth time step,  $T_{i,n}$ (L2/T) is the lateral flow transmissivity,  $h_{i,n}$  (L) is the groundwater table elevation,  $h_{rn}$  (L) is the river water level (which is another boundary condition of the simulation and will be discussed in Sect. 3.2), and  $\Delta x$  (L) is the side length of each model grid.

The variables  $h_{i,n}$ ,  $T_{i,n}$  and  $R_{i,n}$  (i > 0) in Eq. (4) are linked to CLM4.5 as follows. The water table elevation  $h_{i,n}$  is easily obtained by subtracting the groundwater table depth from the ground elevation as:

$$h = h_e - z_{wt}, \tag{5}$$

where  $h_e$  (L) and  $z_{wr}$  (L) are, respectively, the ground elevation and current groundwater table depth of the grid calculated 15 by CLM4.5. To obtain the lateral flow transmissivity  $T_{i,n}$ , we considered two cases in the model. In case A, the groundwater table is within the soil layers of the model (i.e., water table lies within 3.8m from surface-water table depth is deeper than 3.8m) and the transmissivity can be expressed as:

$$T = T_1 + T_2,$$
 (6)

$$T_{1} = \begin{cases} K_{i} \times (z_{h,i} - z_{wi}) + \sum_{j=i+1}^{10} K_{j} \Delta z_{j}, i < 10\\ K_{10} \times (z_{h,10} - z_{wi}), i = 10 \end{cases},$$
(7)

$$T_{2} = \int_{0}^{\infty} K(z') dz' = \int_{0}^{\infty} K_{10} e^{-\frac{z'}{f}} dz' = K_{10} f, \qquad (8)$$

where  $T_{L}$  (L2/T) and  $T_{2}$  (L2/T) are respectively is the lateral flow transmissivity within and outside in the 10th soil layers of CLM4.5,  $T_{2}$  (L2/T) is the transmissivity out of the 10th soil layers, *j* is the number of soil layer denoted by CLM4.5,  $K_{j}$  (L/T) and *f* (L) are, respectively, the lateral hydraulic conductivity of the  $j_{a}^{th}$  soil layer and the e-folding length (which will be discussed later), and  $-\Delta Z_{j}$  (L) is the thickness of the  $j_{a}^{th}$  soil layer, *i* is the soil layer where within which the groundwater table lies-in,  $z_{b,i}$  (L) is the lower boundary depth bottom level of the  $j_{a}^{th}$  soil layer. *i* (in depth), *z'* (L) is the relative depth to the bottom boundary of the 10th soil layer (where z' = z - 3.8, z > 3.8 m), and K(z') (L/T) is the lateral hydraulic conductivity at relative depth  $z'_{a}$ . Based on Fan et al. (2007), we also applied an estimation of the lateral hydraulic

10 conductivity at depth below the  $10_{a}^{th}$  soil layer in Eq. (8) as:

15

$$K(z') = K_{10}e^{-\frac{z}{f}},$$

z'

(9)

where  $K_{40}$  (L/T) is the lateral hydraulic conductivity at the 10th soil layer, z' (L) is the relative depth to the bottom boundary of the 10th soil layer, and K(z') (L/T) is the lateral hydraulic conductivity at relative depth z'. In CLM4.5, only the vertical hydraulic conductivity is provided. So to obtain the lateral hydraulic conductivity  $K_j$  of each soil layer, we applied the assumption of Fan et al. (2007) such that the lateral conductivity is related to the vertical hydraulic conductivity and the content of clay for local soil as:

$$K_j = K_j' \times P_{clay}, \tag{10}$$

where  $K_j$  (L/T) is the vertical hydraulic conductivity provided by CLM4.5 and  $P_{clay}$  is the percentage of clay in local soil, as provided by surface data of CLM4.5. The e-folding length f in Eq. (8) is a parameter representing the local

 帶格式的: 字体: 倾斜

 帶格式的: 字体: 倾斜

 帶格式的: 上标

 帶格式的: 上标

sediment-bedrock profile, which is complex depending on tectonics, weathering and erosion-deposition processes. In this study, we simply implemented an estimation of Fan et al. (2007) to relate e-folding length to terrain slope as:

$$f = \begin{cases} \frac{20}{1+125\beta}, \beta \le 0.16\\ 1, \beta > 0.16 \end{cases}, \tag{11}$$

**5 where $\beta$ (radian) represents the terrain slope and can be obtained from the surface data of CLM4.5.**

In case B, where the groundwater table is positioned below the 10th soil layer of CLM4.5, the  $T_{i,n}$  can be calculated as:

$$T = \int_{z_{wr}-3.8}^{\infty} K(z') dz' = \int_{z_{wr}-3.8}^{\infty} K_{10} e^{-\frac{z'}{f}} dz' = K_{10} f e^{\frac{3.8 - z_{wr}}{f}}_{z_{z}}$$
(12)

where  $z_{110}$  (L) is the lower boundary depth of the 10th soil layer of CLM4.5. We also applied the parameterization of Eq. (9)

in Eq. (12).

20

10 In Eq. (4),  $T_{0,n}$  (L2/T) is the flow transmissivity of the river with respect to groundwater-river exchange. Based on Xie and Yuan (2010), flow transmissivity can be expressed as:

$$T_0 = K_r w, \tag{13}$$

where w (L) is the river width obtained from measured data and  $K_r$  (L2/T) is the hydraulic conductivity at the river bed (which will be discussed in Sect. 3.2). —

15 2.32 Combining the stream-aquifer water interaction scheme with CLM4.5

Generally CLM4.5 is used for large scale simulations (global/continental) using relatively coarse grid resolution (about 0.1 1 degree), and these simulations usually make use of a horizontal-2D 2D lateral grid structure. However, in the investigation of the effects of stream aquifer water interaction over riverbank over riverbank, (especially the intensities of these effects at sites with different distances to river), only the 1D direction which is perpendicular to the river is matter. Furthermore, the spatial scale of for the stream aquifer interaction is usually restricted within several hundreds meters. So to make the CLM4.5 suitable in the 1D and finetiny scale simulation, some modifications and special settings should be

调整西文与
空格 | 定义网
中文之 | 格后不
.间的空 | 调整右
格,不i | 缩进,
调整中 | 无孤行控制。
[文和数字之] | 一不
可的 |
|------------------|-----------------------------|------------|-------------|-------------|------------|-------------------|----------|
* * *
Finally, Finally, tTthe lateral water recharge (or discharge) rate  $R_{i,n}$  in Eq. (4) is linked to CLM4.5 as follows:

$$-\begin{cases} z_{wt\_new} = z_{wt\_ori} - \frac{R \times \Delta t}{s_y}, \\ W_{new} = W_{ori} + R \times \Delta t \end{cases}$$
(14)

- 5 where  $\Delta t$  (T) is the time step of CLM4.5,  $s_y$  is the aquifer specific yield calculated by CLM4.5,  $z_{wt_ori}$  (L) and  $z_{wt_new}$  (L) are, respectively, the original simulated groundwater table depth by CLM4.5 and the updated value after considering the later flow flux, and  $W_{ori}$  (L) and  $W_{new}$  (L) are, respectively, the original simulated aquifer water storage by CLM4.5 and the updated value after considering the lateral flow flux.
- Equations (4) to (14) are applied incorporated in CLM4.5 to renew the values of groundwater table depth and aquifer water storage at every time step. Other hydrological and ecological variables will be in turn be modified by these changes as the model continues to operate. Additionally, the subsurface runoff calculation in the original version of CLM4.5 was turned off and replaced by our lateral flow scheme because in fact groundwater lateral flow was an explicit representation for the subsurface runoff process in the model.

[revised manuscript text omitted]

|-------|-------------------------------------------------------------------|--------------------------------------------------------------------------------------------|------------------------------------------------------------------------------------------------------------------------------------|------------------------------------------------------------------------------------------------------------------------------------|
|       |                                                                   |                                                                                            |                                                                                                                                    |                                                                                                                                    |
|       |                                                                   |                                                                                            |                                                                                                                                    |                                                                                                                                    |
|       |                                                                   |                                                                                            |                                                                                                                                    |                                                                                                                                    |
|       | <li>带格式的:</li><li>带格式的:</li><li>带格式的:</li><li>带格式的:</li> |  <li>带格式的:字体颜色:</li> <li>带格式的:字体颜色:</li> <li>带格式的:字体颜色:</li> <li>带格式的:字体颜色:</li>  |  <li>带格式的:字体颜色:自动设置</li> <li>带格式的:字体颜色:自动设置</li> <li>带格式的:字体颜色:自动设置</li> <li>带格式的:字体颜色:自动设置</li> <li>带格式的:字体颜色:自动设置</li>  |  <li>带格式的:字体颜色:自动设置</li> <li>带格式的:字体颜色:自动设置</li> <li>带格式的:字体颜色:自动设置</li> <li>带格式的:字体颜色:自动设置</li> <li>带格式的:字体颜色:自动设置</li>  |

25

Telemetry Experimental Research (HiWATER, Li et al., 2013; Liu et al., 20145). The observations covered all time periods of our simulations with a time interval of 1800 s0.5 h. First, the TEST and CTL were started from the default initial condition of CLM4.5 (seen in Oleson et al. 2013) and run 700 years under each configuration (with and without stream-aquifer water interaction), cyclically using the atmospheric forcing and observed water level data. Then, the TEST

5 and CTL would start their formal runs from 1 July 2012 to 30 June 2013 using the restart files produced by the former 700-year spin-up.

Both the TEST and CTL runs began from restart files of the 700 year spin-up conducted for each configuration, eyelically using the atmospheric forcing and observed water level data.

4 Results Eco-hydrological effects of stream-aquifer water interaction

**10 4.1 Validation**

First, we validated our model using results from the sensitivity experiments. Both the responses of groundwater table in short-term (within 7 days) and in-long-term (within 160 days) response of groundwater table to river levels  $h_{z}$  and river bed water conductivities  $K_{z}$  simulations were displayed plotted. The results In the case ((a) of) (river recharging groundwater), Figure 4a 4d shows the short-term responses of water table to the different river water levels as the first

15 sensitivity test (described in Sect. 3.2), and Figure 4e 4h shows the short term responses of water tables to the different river bed water conductivities as the second sensitivity test () from of \_\_\_\_\_

of the case (a) (river recharging groundwater) were plotted displayed in the Figure 4. Figure 4a-4h show the time series of the simulated groundwater table depths for each grid cell in the first sensitivity experiment  $Q_{tr}$  was varied and  $K_r$  was held as constant)through the first 7 days (Figure 4a 4d) and 160 days (Figure 4e 4h). From the figures them, we can see that the

- 20 groundwater table depth near the river channel is significantly reduced increased deepened reduced (groundwater head table-is- elevateddeelinedelevated) as the river water level- increasesdecreasesincreases. This is because, as Eq. (1) shows, the higher river water level induces a greater hydraulic gradient, which enhances lateral recharge to the riparian aquifer. This effect is significant in both short-term and long-term simulations, indicating the essential role of river water level in the controlling of riparian groundwater table. Figure 4i-4p show the time series of the simulated groundwater table
- 25 depths for each grid cell in the second experiment ( $h_{t}$  was held as constant and  $K_{t}$  was varied). From the figures, over the

─ 带格式的:字体:倾斜
 ─ 带格式的: 字体:倾斜

short-term simulation (Figure 4i-4l), the effect of  $K_r$  is significant over the short-term simulation (Figure 4i-4l): As the  $K_r$ ranged from 3 m d-1 to 24 m d-1, the time spent by the nearest grid (to the river) to get the equilibrium state is shortened from 2 days to 0.5 days. However, after over the long-term simulation (Figure 4m-4p), the groundwater table depths variations are similar for all values of  $K_{c_{5}}$  indicating that the equilibrium state of ground-ground-water table equilibrium 5 along the river channel is not very sensitive to  $K_z$  compared with  $h_r$ . Thisat is because, river bed water conductivity  $K_z$  only connects the river and the nearest model grid to the river, while the rest of grids (not next to river) are not directly influenced by Kr and more affected by the lateral hydraulic conductivity -K-(in Eq. (9)) of of the- riverbank soil (in Eq. (9))<del>I than the river bed water conductivity K.</del>. The results from case (b) (groundwater recharging river) were plotted in the Figure 5. The conclusions from Figure 5 are similar as Figure 4: River level is matter over both short-term and long-term 10 simulations in the controlling of riparian water table, while the while river bed water conductivity is more more important in the thecontrolling of the short-term water table variation than the controlling of long-term water table equilibrium. Figure 4 and Figure 5 jointly validated that our model could reasonably reproduce both the processes of river recharging groundwater -as well as the opposite process oandf groundwater recharging river. short-term response of water table to the river level than the equilibrium state of water table after a long-term simulation. .

[revised manuscript text omitted]

We used relative temperature of the nearest grid to the stream to emphasize spatial variability. The soil and ground and soil heat transfer algorithm in CLM is applied on the vertical direction as:

|--------------------------------------------------------------------------------------------------------------------------------------------------------------------------|---|----------------------|
| $\frac{\partial z}{\partial t} = \frac{\partial z}{\partial z} \begin{bmatrix} \lambda & \partial z \end{bmatrix}, \qquad (15)$                                          |   |                      |
| where $\tau$ (m) is in the vertical direction and is positive downward. $T(K)$ is the temperature $-c(Im^3 K^4)$ is the volumetric                                       |   | 带格式的: 字体:倾斜   |
|                                                                                                                                                                          |   | ( 带格式的: 字体:倾斜 |
| soil heat soil capacity, $\lambda$ (W m + -K + ) is the thermal conductivity and $t$ (s) is time. Both the thermal properties of $c$ and $\lambda$ |   | ( 带格式的: 字体:倾斜 |
|                                                                                                                                                                          |   | ( 带格式的: 字体:倾斜 |
| depend on the soft water content as follow (assuming no soft fee for concise expression):                                                                                |   |                      |
| $c = c \left(1 - \theta\right) + c \theta \qquad (16)$                                                                                                                   | / | (域代码已更改              |
|                                                                                                                                                                          |   |                      |
| $\lambda = K_{a}\lambda_{corr} + (1 - K_{a})\lambda_{torr}.$ (17)                                                                                                        |   | 域代码已更改               |
|                                                                                                                                                                          |   |                      |
| $K_e = \lg \left( \frac{-\iota_q}{\theta_{sat}} \right) + 1, $ (18)                                                                                                      |   |                      |

[revised manuscript text omitted]
 | Name      | Latitude       | Longitude       | Width | Riverbank | Bottom    | Flow      |
|-----------|-----------|----------------|-----------------|-------|-----------|-----------|-----------|
| section   |           |                |                 | (m)   | elevation | elevation | direction |
|           |           |                |                 |       | (m)       | (m)       |           |
| 1         | 213       | 38 °54'43.55"N | 100 °20'41.05"E | 330   | 1493.1    | 1488.8    | Northeast |
|           | Bridge    |                |                 |       |           |           |           |
| 2         | 312       | 38°59′51.71″N  | 100°24′38.76″E  | 70    | 1402      | 1397      | Northeast |
|           | Bridge    |                |                 |       |           |           |           |
| 3         | Tielu     | 39 °2'33.08"N  | 100 25'49.42"E  | 50    | 1382      | 1379.25   | Northeast |
|           | Bridge    |                |                 |       |           |           |           |
| 4         | Pingchuan | 39 °20'2.03"N  | 100 °5'49.63"E  | 130   | 1323.8    | 1319      | West      |
|           | Bridge    |                |                 |       |           |           |           |
| 5         | Gaotai    | 39 °23'22.93"N | 99 49'37.29"E   | 210   | 1295.5    | 1288.5    | West      |
|           | Bridge    |                |                 |       |           |           |           |

Table 1 The locations and relevant information about the five selected sections used in simulations.

| Table 2 The soil types and vegetation types over both sides of the five selected sections used in simulations, |                                   |                  |                   |                        |                         |
|----------------------------------------------------------------------------------------------------------------|-----------------------------------|------------------|-------------------|------------------------|-------------------------|
| Number of section                                                                                              | Name                              | Soil type (Left) | Soil type (Right) | Vegetation type (Left) | Vegetation type (Right) |
| 1                                                                                                              | 213 Bridge                        | Sand             | Silt              | Bare ground            | Corn                    |
| 2                                                                                                              | 312 Bridge                        | Silt             | Silt              | Corn                   | Grass                   |
| 3                                                                                                       | Tielu Bridge                      | Silt             | Silt              | Corn                   | Grass and corn          |
| 4                                                                                                       | Pingchuan
Bridge | Silt             | Silt              | Grass and Corn         | Grass and corn          |
| 5                                                                                                       | Gaotai Bridge                     | Sand             | Sand              | Grass                  | Corn                    |

号                 | 左, 行距: 单倍行距, 孤行控制, 不取消行                                                             |
|-------------|-----------------------------------|-------------------------------------------------------------------------------------|
字符,定义
孤行控制,
和数字之间 | 居中, 缩进: 左侧: 0 厘米, 首行缩进: 0
网格后不调整右缩进, 行距: 2 倍行距, 无
不调整西文与中文之间的空格, 不调整中文
的空格 |
